# Smooth Interactive Submodular Set Cover

**Bryan He**
Stanford University
bryanhe@stanford.edu

**Yisong Yue**
California Institute of Technology
yyue@caltech.edu

## Abstract

Interactive submodular set cover is an interactive variant of submodular set cover over a hypothesis class of submodular functions, where the goal is to satisfy all sufficiently plausible submodular functions to a target threshold using as few (cost-weighted) actions as possible. It models settings where there is uncertainty regarding which submodular function to optimize. In this paper, we propose a new extension, which we call *smooth interactive submodular set cover*, that allows the target threshold to vary depending on the plausibility of each hypothesis. We present the first algorithm for this more general setting with theoretical guarantees on optimality. We further show how to extend our approach to deal with real-valued functions, which yields new theoretical results for real-valued submodular set cover for both the interactive and non-interactive settings.

## 1 Introduction

In interactive submodular set cover (ISSC) [10, 11, 9], the goal is to interactively satisfy all plausible submodular functions in as few actions as possible. ISSC is a wide-encompassing framework that generalizes both submodular set cover [24] by virtue of being interactive, as well as some instances of active learning by virtue of many active learning criteria being submodular [12, 9].

A key characteristic of ISSC is the a priori uncertainty regarding the correct submodular function to optimize. For example, in personalized recommender systems, the system does not know the user's preferences a priori, but can learn them interactively via user feedback. Thus, any algorithm must choose actions in order to disambiguate between competing hypotheses as well as optimize for the most plausible ones – this issue is also known as the exploration-exploitation tradeoff.

In this paper, we propose the *smooth interactive submodular set cover* problem, which addresses two important limitations of previous work. The first limitation is that conventional ISSC [10, 11, 9] only allows for a single threshold to satisfy, and this "all or nothing" nature can be inflexible for settings where the covering goal should vary smoothly (e.g., based on plausibility). In smooth ISSC, one can smoothly vary the target threshold of the candidate submodular functions according to their plausibility. In other words, the less plausible a hypothesis is, the less we emphasize maximizing its associated utility function. We present a simple greedy algorithm for smooth ISSC with provable guarantees on optimality. We also show that our smooth ISSC framework and algorithm fully generalize previous instances of and algorithms for ISSC by reducing back to just one threshold.

One consequence of smooth ISSC is the need to optimize for real-valued functions, which leads to the second limitation of previous work. Many natural classes of submodular functions are real-valued (cf. [25, 5, 17, 21]). However, submodular set cover (both interactive and non-interactive) has only been rigorously studied for integral or rational functions with fixed denominator, which highlights a significant gap between theory and practice. We propose a relaxed version of smooth ISSC using an approximation tolerance $\epsilon$, such that one needs only to satisfy the set cover criterion to within $\epsilon$. We extend our greedy algorithm to provably optimize for real-valued submodular functions within this $\epsilon$ tolerance. To the best of our knowledge, this yields the first theoretically rigorous algorithm for real-valued submodular set cover (both interactive and non-interactive).

---
**Problem 1** Smooth Interactive Submodular Set Cover
---
1: **Given:**
  1. Hypothesis class $H$ (does not necessarily contain $h^*$)
  2. Query set $\mathcal{Q}$ and response set $\mathcal{R}$ with known $q(h) \subseteq \mathcal{R}$ for $q \in \mathcal{Q}, h \in H$
  3. Modular query cost function $c$ defined over $\mathcal{Q}$
  4. Monotone submodular objective functions $F_h : 2^{\mathcal{Q} \times \mathcal{R}} \to \mathbb{R}_{\geq 0}$ for $h \in H$
  5. Monotone submodular distance functions $G_h : 2^{\mathcal{Q} \times \mathcal{R}} \to \mathbb{R}_{\geq 0}$ for $h \in H$, with $G_h(S \oplus (q,r)) - G_h(S) = 0$ for any $S$ if $r \in q(h)$
  6. Threshold function $\alpha : \mathbb{R}_{\geq 0} \to \mathbb{R}_{\geq 0}$ mapping a distance to required objective function value
2: **Protocol:** For $i = 1, \ldots, \infty$: ask a question $\hat{q}_i \in \mathcal{Q}$ and receive a response $\hat{r}_i \in \hat{q}_i(h^*)$.
3: **Goal:** Using minimal cost $\sum_i c(\hat{q}_i)$, terminate when $F_h(\hat{S}) \geq \alpha(G_h(S^*))$ for all $h \in H$, where $\hat{S} = \{(\hat{q}_i, \hat{r}_i)\}_i$ and $S^* \triangleq \bigcup_{q \in \mathcal{Q}, r \in q(h^*)} \{(q,r)\}$.

---

## 2  Background

**Submodular Set Cover.** In the basic submodular set cover problem [24], we are given an action set $\mathcal{Q}$ and a monotone submodular set function $F : 2^{\mathcal{Q}} \to \mathbb{R}_{\geq 0}$ that maps subsets $A \subseteq \mathcal{Q}$ to non-negative scalar values. A set function $F$ is monotone and submodular if and only if:

$$\forall A \subseteq B \subseteq \mathcal{Q}, q \in \mathcal{Q}: \quad F(A \oplus q) \geq F(A) \quad \text{and} \quad F(A \oplus q) - F(A) \geq F(B \oplus q) - F(B),$$

respectively, where $\oplus$ denotes set addition (i.e., $A \oplus q \equiv A \cup \{q\}$). In other words, monotonicity implies that adding a set always yields non-negative gain, and submodularity implies that adding to a smaller set $A$ results in a larger gain than adding to a larger set $B$. We also assume that $F(\emptyset) = 0$.

Each $q \in \mathcal{Q}$ is associated with a modular or additive cost $c(q)$. Given a target threshold $\alpha$, the goal is to select a set $A$ that satisfies $F(A) \geq \alpha$ with minimal cost $c(A) = \sum_{q \in A} c(q)$. This problem is NP-hard; but for integer-valued $F$, simple greedy forward selection can provably achieve near-optimal cost of at most $(1 + \ln(\max_{a \in \mathcal{Q}} F(\{a\})))OPT$ [24], and is typically very effective in practice.

One motivating application is content recommendation [5, 4, 25, 11, 21], where $\mathcal{Q}$ are items to recommend, $F(A)$ captures the utility of $A \subseteq \mathcal{Q}$, and $\alpha$ is the satisfaction goal. Monotonicity of $F$ captures the property that total utility never decreases as one recommends more items, and submodularity captures the the diminishing returns property when recommending redundant items.

**Interactive Submodular Set Cover.** In the basic interactive setting [10], the decision maker must optimize over a hypothesis class $H$ of submodular functions $F_h$. The setting is interactive, whereby the decision maker chooses an action (or query) $q \in Q$, and the environment provides a response $r \in \mathcal{R}$. Each query $q$ is now a function mapping hypotheses $H$ to responses $\mathcal{R}$ (i.e., $q(h) \in R$), and the environment provides responses according to an unknown true hypothesis $h^* \in H$ (i.e., $r \equiv q(h^*)$). This process iterates until $F_{h^*}(S) \geq \alpha$, where $S$ denotes the set of observed question/response pairs: $S = \{(q,r)\} \subseteq \mathcal{Q} \times \mathcal{R}$. The goal is to satisfy $F_{h^*}(S) \geq \alpha$ with minimal cost $c(S) = \sum_{(q,r) \in S} c(q)$.

For example, when recommending movies to a new user with unknown interests (cf. [10, 11]), $H$ can be a set of user types or movie genres (e.g., $H = \{\text{Action}, \text{Drama}, \text{Horror}, \ldots\}$). Then $\mathcal{Q}$ would contain individual movies that can be recommended, and $\mathcal{R}$ would be a "yes" or "no" response or an integer rating representing how interested the user (modeled as $h^*$) is in a given movie.

The interactive setting is both a learning and covering problem, as opposed to just a covering problem. The decision maker must balance between disambiguating between hypotheses in $H$ (i.e., identifying which is the true $h^*$) and satisfying the covering goal $F_{h^*}(S) \geq \alpha$; this issue is also known as the exploration-exploitation tradeoff. Noisy ISSC [11] extends basic ISSC by no longer assuming the true $h^*$ is in $H$, and uses a distance function $G_h$ and tolerance $\kappa$ such that the goal is to satisfy $F_h(S) \geq \alpha$ for all sufficiently plausible $h$, where plausibility is defined as $G_h(S) \leq \kappa$.

## 3  Problem Statement

We now present the *smooth interactive submodular set cover* problem, which generalizes basic and noisy ISSC [10, 11] (described in Section 2). Like basic ISSC, each hypothesis $h \in H$ is associated with a utility function $F_h : 2^{\mathcal{Q} \times \mathcal{R}} \to \mathbb{R}_{\geq 0}$ that maps sets of query/response pairs to

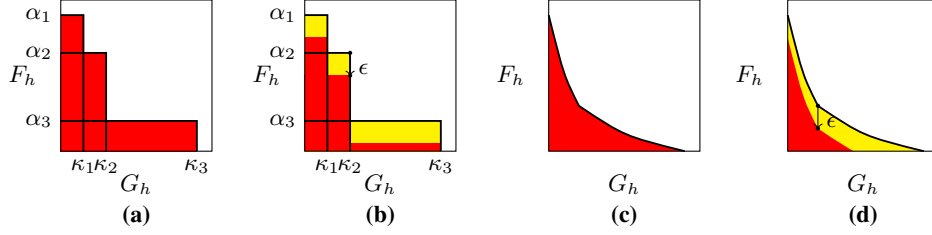

Figure 1: Examples of (a) multiple thresholds, (b) approximate multiple thresholds, (c) a continuous convex threshold, and (d) an approximate continuous convex threshold. For the approximate setting, we essentially allow for satisfying any threshold function that resides in the yellow region.

non-negative scalars. Like noisy ISSC, the hypothesis class $H$ does not necessarily contain the true $h^*$ (i.e., the agnostic setting). Each $h \in H$ is associated with a distance or disagreement function $G_h : 2^{\mathcal{Q} \times \mathcal{R}} \to \mathbb{R}_{\geq 0}$ which maps sets of question/response pairs to a disagreement score (i.e., the larger $G_h(S)$ is, the more $h$ disagrees with $S$). We further require that $F_h(\emptyset) = 0$ and $G_h(\emptyset) = 0$.

Problem 1 describes the general problem setting. Let $S^* \triangleq \bigcup_{q \in Q, r \in q(h^*)} \{(q,r)\}$ denote the set of all possible question/responses pairs given by $h^*$. The goal is to construct a question/response set $\hat{S}$ with minimal cost such that, for every $h \in H$ we have $F_h(\hat{S}) \geq \alpha(G_h(S^*))$, where $\alpha(\cdot)$ maps disagreement values to desired utilities. In general, $\alpha(\cdot)$ is a non-increasing function, since the goal is to optimize more the most plausible hypotheses in $H$. We describe two versions of $\alpha(\cdot)$ below.

**Version 1: Step Function (Multiple Thresholds).** The first version uses a decreasing step function (see Figure 1(a)). Given a pair of sequences $\alpha_1 > \ldots > \alpha_N > 0$ and $0 < \kappa_1 < \ldots < \kappa_N$, the threshold function is $\alpha(v) = \alpha_{n_\kappa(v)}$ where $n_\kappa(v) = \min\{n \in \{0, \ldots, N+1\} | v < \kappa_n\}$, and $\alpha_0 \triangleq \infty, \alpha_{N+1} \triangleq 0, \kappa_0 \triangleq 0, \kappa_{N+1} \triangleq \infty$. The goal in Problem 1 is equivalently: "$\forall h \in H$ and $n = 1, \ldots, N$: satisfy $F_h(\hat{S}) \geq \alpha_n$ whenever $G_h(S^*) < \kappa_n$." This version is a strict generalization of noisy ISSC, which uses only a single $\alpha$ and $\kappa$.

**Version 2: Convex Threshold Curve.** The second version uses a convex $\alpha(\cdot)$ that decreases continuously as $G_h(S^*)$ increases (see Figure 1(c)), and is not a strict generalization of noisy ISSC.

**Approximate Thresholds.** Finally, we also consider a relaxed version of smooth ISSC, whereby we only require that the objectives $F_h$ be satisfied to within some tolerance $\epsilon \geq 0$. More formally, we say that we approximately solve Problem 1 with tolerance $\epsilon$ if its goal is redefined as: "*using minimal cost, $\sum_i c(\hat{q}_i)$, guarantee $F_h(\hat{S}) \geq \alpha(G_h(S^*)) - \epsilon$ for all $h \in H$.*" See Figure 1(b) & 1(d) for the approximate versions of the multiple tresholds and convex versions, respectively.

ISSC has only been rigorously studied when the utility functions are $F_h$ are rational-valued with a fixed denominator. We show in Section 4.3 how to efficiently solve the approximate version of smooth ISSC when $F_h$ are real-valued, which also yields a new approach for approximately solving the classical non-interactive submodular set cover problem with real-valued objective functions.

# 4 Algorithm & Main Results

A key question in the study of interactive optimization is how to balance the exploration-exploitation tradeoff. On the one hand, one should exploit current knowledge to efficiently satisfy the plausible submodular functions. However, hypotheses that seem plausible might actually not be due to imperfections in the algorithm's knowledge. One should thus explore by playing actions that disambiguate the plausibility of competing hypotheses. Our setting is further complicated due to also solving a combinatorial optimization problem (submodular set cover), which is in general intractable.

## 4.1 Approach Outline

We present a general greedy algorithm, described in Algorithm 1 below, for solving smooth ISSC with provably near-optimal cost. Algorithm 1 requires as input a submodular meta-objective $\bar{F}$

---
**Algorithm 1** Worst Case Greedy Algorithm for Smooth Interactive Submodular Set Cover
---
1: **input:** $\bar{F}$                                                   *// Submodular Meta-Objective*
2: **input:** $\bar{F}_{max}$                                 *// Termination Threshold for $\bar{F}$*
3: **input:** $\mathcal{Q}$                                                *// Query or Action Set*
4: **input:** $\mathcal{R}$                                                  *// Response Set*
5: $S \leftarrow \emptyset$
6: **while** $\bar{F}(S) < \bar{F}_{max}$ **do**
7:     $\hat{q} \leftarrow \operatorname{argmax}_{q \in \mathcal{Q}} \min_{r \in \mathcal{R}} \left( \bar{F}(S \oplus (q,r)) - \bar{F}(S) \right) / c(q)$
8:     Play $\hat{q}$, observe $\hat{r}$
9:     $S \leftarrow S \oplus (\hat{q}, \hat{r})$
10: **end while**
---

| Variable | Definition |
|---|---|
| $H$ | Set of hypotheses |
| $\mathcal{Q}$ | Set of actions or queries |
| $\mathcal{R}$ | Set of responses |
| $F_h$ | Monotone non-decreasing submodular utility function |
| $G_h$ | Monotone non-decreasing submodular distance function |
| $\bar{F}$ | Monotone non-decreasing submodular function unifying $F_h$, $G_h$ and the thresholds |
| $\bar{F}_{max}$ | Maximum value held by $\bar{F}$ |
| $D_F$ | Denominator for $F_h$ (when rational) |
| $D_G$ | Denominator for $G_h$ (when rational) |
| $\alpha(\cdot)$ | Continuous convex threshold |
| $\alpha_i$ | Thresholds for $F$ ($\alpha_1$ is largest) |
| $\kappa_i$ | Thresholds for $G$ ($\kappa_1$ is smallest) |
| $N$ | Number of thresholds |
| $\epsilon$ | Approximation tolerance for the real-valued case |
| $F'_h$ | Surrogate utility function for the approximate version |
| $\alpha'_n$ | Surrogate thresholds for the approximate version |

Figure 2: Summary of notation used. The top portion is used in all settings. The middle portion is used for the multiple thresholds setting. The bottom portion is used for real-valued functions.

that quantifies the exploration-exploitation trade-off, and the specific instantiation of $\bar{F}$ depends on which version of smooth ISSC is being solved. Algorithm 1 greedily optimizes for the worst case outcome at each iteration (Line 7) until a termination condition $\bar{F} \geq \bar{F}_{max}$ has been met (Line 6).

The construction of $\bar{F}$ is essentially a reduction of smooth ISSC to a simpler submodular set cover problem, and generalizes the reduction approach in [11]. In particular, we first lift the analysis of [11] to deal with multiple thresholds (Section 4.2). We then show how to deal with approximate thresholds in the real-valued setting (Section 4.3), which finally allows us to address the continuous threshold setting (Section 4.4). Our cost guarantees are stated relative to the *general cover cost* (GCC), which lower bounds the optimal cost, as stated in Definition 4.1 and Lemma 4.2 below. Via this reduction, we can show that our approach achieves cost bounded by $(1 + \ln \bar{F}_{max})GCC \leq (1 + \ln \bar{F}_{max})OPT$. For clarity of exposition, all proofs are deferred to the supplementary material.

**Definition 4.1** (General Cover Cost (GCC)). *Define oracles* $T \in R^Q$ *to be functions mapping questions to responses and* $T(\hat{Q}) \triangleq \bigcup_{\hat{q}_i \in \hat{Q}} \{(\hat{q}_i, T(\hat{q}_i))\}$. $T(\hat{Q})$ *is the set of question-response pairs given by* $T$ *for the set of questions* $\hat{Q}$. *Define the General Cover Cost as:*

$$GCC \triangleq \max_{T \in R^Q} \left( \min_{\hat{Q}: \bar{F}(T(\hat{Q})) \geq \bar{F}_{max}} c(\hat{Q}) \right).$$

**Lemma 4.2** (Lemma 3 from [11]). *If there is a question asking strategy for satisfying* $\bar{F}(\hat{S}) \geq \bar{F}_{max}$ *with worst case cost* $C^*$, *then* $GCC \leq C^*$. *Thus* $GCC \leq OPT$.

## 4.2 Multiple Thresholds Version

We begin with the multiple thresholds version. In this section, we assume that each $F_h$ and $G_h$ are rational-valued with fixed denominators $D_F$ and $D_G$, respectively.[1] We first define a doubly

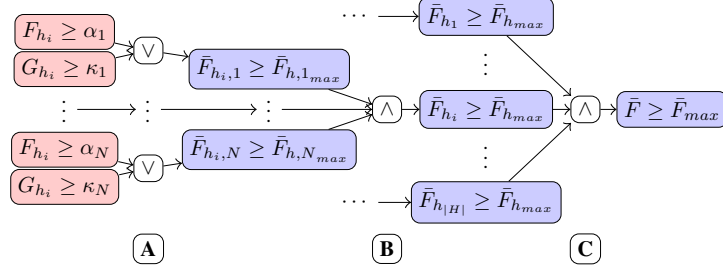

$$\boxed{A} \qquad\qquad \boxed{B} \qquad\qquad \boxed{C}$$

Figure 3: Depicting the relationship between the terms defined in Definition 4.3. (A) If $\bar{F}_{h_i,n} \geq \bar{F}_{h_i,n_{max}} = (\alpha_n - \alpha_{n+1})(\kappa_n - \kappa_{n-1})$, then either $F_{h_i} \geq \alpha_n$ or $G_{h_i} \geq \kappa_n$; this generates the tradeoff between satisfying the either of the two thresholds. (B) If $\bar{F}_{h_i} \geq \bar{F}_{h_{max}}$, then $\bar{F}_{h_i,n} \geq \bar{F}_{h_i,n_{max}}$ $\forall i \in \{1, \dots, N\}$; this enforces that all $i$, at least one of the thresholds $\alpha_i$ or $\kappa_i$ must be satisfied. (C) If $\bar{F} \geq \bar{F}_{max}$, then $\bar{F}_h \geq \bar{F}_{h_{max}}$ $\forall h \in H$; this enforces that all hypotheses must be satisfied.

truncated version of each hypothesis submodular utility and distance function:

$$F_{h,\alpha_n,\alpha_j}(\hat{S}) \stackrel{\triangle}{=} \max(\min(F_h(\hat{S}), \alpha_n), \alpha_j) - \alpha_j, \tag{1}$$

$$G_{h,\kappa_n,\kappa_j}(\hat{S}) \stackrel{\triangle}{=} \max(\min(G_h(\hat{S}), \kappa_n), \kappa_j) - \kappa_j. \tag{2}$$

In other words, $F_{h,\alpha_n,\alpha_j}$ is truncated from below at $\alpha_j$ and from above at $\alpha_n$ (it is assumed that $\alpha_n > \alpha_j$), and is offset by $-\alpha_j$ so that $F_{h,\alpha_n,\alpha_j}(\emptyset) = 0$. $G_{h,\kappa_n,\kappa_j}$ is constructed analogously. Using (1) and (2), we can define the general forms of $\bar{F}$ and $\bar{F}_{max}$, which can be instantiated to address different versions of smooth ISSC.

**Definition 4.3** (General form of $\bar{F}$ and $\bar{F}_{max}$).

$$\bar{F}_{h,n}(\hat{S}) \stackrel{\triangle}{=} \left( (\kappa_n - \kappa_{n-1}) - G_{h,\kappa_n,\kappa_{n-1}}(\hat{S}) \right) F_{h,\alpha_n,\alpha_{n+1}}(\hat{S}) + G_{h,\kappa_n,\kappa_{n-1}}(\hat{S})(\alpha_n - \alpha_{n+1}),$$

$$\bar{F}_h(\hat{S}) \stackrel{\triangle}{=} C_{\bar{F}} \sum_{n=1}^{N} \left[ \left( \prod_{j \neq n} (\kappa_j - \kappa_{j-1}) \right) \bar{F}_{h,n}(\hat{S}) \right],$$

$$\bar{F}(\hat{S}) \stackrel{\triangle}{=} \sum_{h \in H} \bar{F}_h(\hat{S}), \ \ \bar{F}_{max} \stackrel{\triangle}{=} |H| C_F C_G.$$

The coefficient $C_{\bar{F}}$ converts each $\bar{F}_h$ to be integer-valued, $C_F$ is the contribution to $\bar{F}_{max}$ from $F_h$ and $\alpha_n$, and $C_G$ is the contribution to $\bar{F}_{max}$ from $G_h$ and $\kappa_n$.

**Definition 4.4** (Multiple Thresholds Version of ISSC). *Given $\alpha_1, \dots, \alpha_N$ and $\kappa_1, \dots, \kappa_N$, we instantiate $\bar{F}$ and $\bar{F}_{max}$ in Definition 4.3 via:*

$$C_{\bar{F}} = D_F D_G^N, \qquad C_F = D_F \alpha_1, \qquad C_G = D_G^N \prod_{n=1}^{N} (\kappa_n - \kappa_{n-1}).$$

$\bar{F}$ in Definition 4.4 trades off between exploitation (maximizing the plausible $F_h$'s) and exploration (disambiguating plausibility in $F_h$'s) by allowing each $\bar{F}_h$ to reach its maximum by either $F_h$ reaching $\alpha_i$ or $G_h$ reaching $\kappa_i$. In other words, each $\bar{F}_h$ can be satisfied with either a sufficiently large utility $F_h$ or large distance $G_h$. Figure 3 shows the logical relationships between these components.

We prove in Appendix A that $\bar{F}$ is monotone submodular, and that finding an $S$ such that $\bar{F}(S) \geq \bar{F}_{max}$ is equivalent to solving Problem 1. For $\bar{F}$ to be submodular, we also require Condition 4.5, which is essentially a discrete analogue to the condition that a continuous $\alpha(\cdot)$ should be convex.

**Condition 4.5.** *The sequence $\langle \frac{\alpha_n - \alpha_{n+1}}{\kappa_n - \kappa_{n-1}} \rangle_{n=1}^{N}$ is non-increasing.*

**Theorem 4.6.** *Given Condition 4.5, Algorithm 1 using Definition 4.4 solves the multiple thresholds version of Problem 1 using cost at most $\left( 1 + \ln \left( |H| D_F D_G^N \alpha_1 \prod_{n=1}^{N} (\kappa_n - \kappa_{n-1}) \right) \right) GCC.$*

If each $G_h$ is integral and $\kappa_n = \kappa_{n-1} + 1$, then the bound simplifies to $(1 + \ln(|H| D_F \alpha_1)) GCC$. We present an alternative formulation in Appendix D.2 that has better bounds when $D_G$ is large, but is less flexible and cannot be easily extended to the real-valued and convex threshold curve settings.

## 4.3 Approximate Thresholds for Real-Valued Functions

Solving even non-interactive submodular set cover is extremely challenging when the utility functions $F_h$ are real-valued. For example, Appendix B.1 describes a setting where the greedy algorithm performs arbitrarily poorly. We now extend the results from Section 4.2 to real-valued $F_h$ and $\alpha_1, \ldots, \alpha_N$.

Rather than trying to solve the problem exactly, we instead solve a relaxed or approximate version, which will be useful for the convex threshold curve setting. Let $\epsilon > 0$ denote a pre-specified approximation tolerance for $F_h$, $\lceil \cdot \rceil_\gamma$ denote rounding up to the nearest multiple of $\gamma$, and $\lfloor \cdot \rfloor_\gamma$ denote rounding down to the nearest multiple of $\gamma$. We define a surrogate problem:

**Definition 4.7** (Approximate Thresholds for Real-Valued Functions). *Define the following approximations to $F_h$ and $\alpha_n$:*

$$F_h'(\hat{S}) \triangleq \frac{D}{\epsilon} \left\lceil F_h(\hat{S}) + \frac{\epsilon}{D} \sum_{i=1}^{|\hat{S}|} (|\mathcal{Q}| + 1 - i) \right\rceil_{\frac{\epsilon}{D}},$$

$$\alpha_n' \triangleq \frac{D}{\epsilon} \left\lfloor \alpha_n - \frac{\epsilon}{D} \sum_{i=1}^{n} \left[ (2N - 2i) D_G^{N-i+1} \prod_{j=i}^{N} (\kappa_j - \kappa_{j-1}) \right] \right\rfloor_{\frac{\epsilon}{D}}$$

$$D \triangleq \left\lceil \sum_{i=1}^{|\mathcal{Q}|} (|\mathcal{Q}| + 1 - i) + \sum_{i=1}^{N} \left[ (2N - 2i) D_G^{N-i+1} \prod_{j=i}^{N} (\kappa_j - \kappa_{j-1}) \right] + 2 \right\rceil$$

*Instantiate $\bar{F}$ and $\bar{F}_{max}$ in Definition 4.3 using $F_h'$, $\alpha_n'$ above, $G_h$, $\kappa_n$ and:*

$$C_{\bar{F}} = D_G^N, \quad C_F = \alpha_1', \quad C_G = D_G^N \prod_{n=1}^{N} (\kappa_n - \kappa_{n-1}).$$

We prove in Appendix B that Definition 4.7 is an instance of a smooth ISSC problem, and that solving Definition 4.7 will approximately solve the original real-valued smooth ISSC problem.

**Theorem 4.8.** *Given Condition 4.5, Algorithm 1 using Definition 4.7 will approximately solve the real-valued multiple thresholds version of Problem 1 with tolerance $\epsilon$ using cost at most $\left( 1 + \ln \left( |H| \alpha_1' D_G^N \prod_{n=1}^{N} (\kappa_n - \kappa_{n-1}) \right) \right) GCC.*

We show in Appendix B.2 how to apply this result to approximately solve the basic submodular set cover problem with real-valued objectives. Note that if $\epsilon$ is selected as the smallest distinct difference between values in $F_h$, then the approximation will be exact.

## 4.4 Convex Threshold Curve Version

We now address the setting where the threshold curve $\alpha(\cdot)$ is continuous and convex. We again solve the approximate version, since the threshold curve $\alpha(\cdot)$ is necessarily real-valued. Let $\epsilon > 0$ be the pre-specified tolerance for $F_h'$. Let $N$ be defined so that $ND_G$ is the maximal value of $G_h$. We convert the continuous version $\alpha(\cdot)$ to a multiple threshold version (with $N$ thresholds) that is within an $\epsilon$-approximation of the former, as shown below.

**Definition 4.9** (Equivalent Multiple Thresholds for Continuous Convex Curve). *Instantiate $\bar{F}$ and $\bar{F}_{max}$ in Definition 4.3 using $G_h$ without modification, and a sequence of thresholds:*

$$F_h'(\hat{S}) \triangleq \frac{D}{\epsilon} \left\lceil F_h(\hat{S}) + \frac{\epsilon}{D} \sum_{i=1}^{|\hat{S}|} (|\mathcal{Q}| + 1 - i) \right\rceil_{\frac{\epsilon}{D}},$$

$$\alpha_n' \triangleq \frac{D}{\epsilon} \left\lfloor \alpha(n) - \frac{\epsilon}{D} \sum_{i=1}^{n} \left[ (2N - 2i) D_G^{N-i+1} \prod_{j=i}^{N} (\kappa_j - \kappa_{j-1}) \right] \right\rfloor_{\frac{\epsilon}{D}}$$

$$\kappa_n \triangleq D_G n$$

*with constants set as:*

$$C_{\bar{F}} = 1, \quad C_F = \alpha'_1, \quad C_G = D_G^N \prod_{n=1}^{N} (\kappa_n - \kappa_{n-1}) = D_G^N.$$

Note that the $F'_h$ are not too expensive to compute. We prove in Appendix C that satisfying this set of thresholds is equivalent to satisfying the original curve $\alpha(\cdot)$ within $\epsilon$-error. Note also that Definition 4.9 uses the same form as Definition 4.7 to handle the approximation of real-valued functions.

**Theorem 4.10.** *Applying Algorithm 1 using Definition 4.9 approximately solves the convex threshold version of Problem 1 with tolerance $\epsilon$ using cost at most:* $\left(1 + \ln\left(|H|\alpha'_1 D_G^N\right)\right) GCC.$

Note that if $\epsilon$ is sufficiently large, then $N$ could in principle be smaller, which can lead to less conservative approximations. There may also be more precise approximations by reducing to other formulations for the multi-threshold setting (e.g., Appendix D.2).

## 5 Simulation Experiments

**Comparison of Methods to Solve Multiple Thresholds.** We compared our multiple threshold method against multiple baselines (see Appendix D for more details) in a range of simulation settings (see Appendix E.1). Figure 4 shows the results. We see that our approach is consistently amongst the best performing methods. The primary competitor is the circuit of constraints approach from [11] (see Appendix D.3 for a comparison of the theoretical guarantees). We also note that all approaches dramatically outperform their worst-case guarantees.

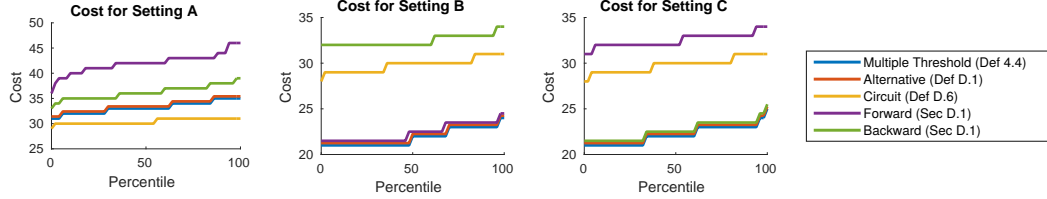

Figure 4: Comparison against baselines in three simulation settings.

**Validating Approximation Tolerances.** We also validated the efficacy of our approximate thresholds relaxation (see Appendix E.2 for more details of the setup). Figure 5 shows the results. We see that the actual deviation from the original smooth ISSC problem is much smaller than the specified $\epsilon$, which suggests that our guarantees are rather conservative. For instance, at $\epsilon = 15$, the algorithm is allowed to terminate immediately. We also see that the cost to completion steadily decreases as $\epsilon$ increases, which agrees with our theoretical results.

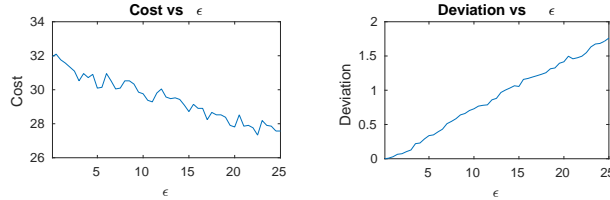

Figure 5: Comparing cost and deviation from the exact function for varying $\epsilon$.

## 6 Summary of Results & Discussion

Figure 6 summarizes the size of $\bar{F}_{max}$ (or $\bar{F}'_{max}$ for real-valued functions) for the various settings. Recall that our cost guarantees take the form $(1 + \ln \bar{F}_{max})OPT$. When $F_h$ are real-valued, then we instead solve the smooth ISSC problem approximately with cost guarantee $(1 + \ln \bar{F}'_{max})OPT$.

Our results are well developed for many different versions of the utility functions $F_h$, but are less flexible for the distance functions $G_h$. For example, even for rational-valued $G_h$, $\bar{F}_{max}$ scales as $D_G^N$, which is not desirable. The restriction of $G_h$ to be rational (or integral) leads to a relatively straightforward reduction of the continuous convex version of $\alpha(\cdot)$ to a multiple thresholds version.

In fact, our formulation can be extended to deal with real-valued $G_h$ and $\kappa_n$ in the multiple thresholds version; however the resulting $\bar{F}$ is no longer guaranteed to be submodular. It is possible that a different assumption than the one imposed in Condition 4.5 is required to prove more general results.

| $F$ | $G$ | Multiple Thresholds | Convex Threshold Curve |
|---|---|---|---|
| Rational | Rational | $\|H\|\alpha_1 D_F D_G^N \prod_{i=1}^N (\kappa_i - \kappa_{i-1})$ | $\|H\|\alpha_1 D_F D_G^N$ |
| Real | Rational | $\|H\|\alpha_1' D_G^N \prod_{i=1}^N (\kappa_i - \kappa_{i-1})$ | $\|H\|\alpha_1' D_G^N$ |

Figure 6: Summarizing $\bar{F}_{max}$. When $F_h$ are real-valued, we show $\bar{F}'_{max}$ instead.

Our analysis appears to be overly conservative for many settings. For instance, all the approaches we evaluated empirically achieved much better performance than their worst-case guarantees. It would be interesting to identify ways to constrain the problem and develop tighter theoretical guarantees.

# 7 Other Related Work

Submodular optimization is an important problem that arises across many settings, including sensor placements [16, 15], summarization [26, 17, 23], inferring latent influence networks [8], diversified recommender systems [5, 4, 25, 21], and multiple solution prediction [1, 3, 22, 19]. However, the majority of previous work has focused on offline submodular optimization whereby the submodular function to be optimized is fixed a priori (i.e., does not vary depending on feedback).

There are two typical ways that a submodular optimization problem can be made interactive. The first is in online submodular optimization, where an unknown submodular function must be re-optimized repeatedly over many sessions in an online or repeated-games fashion [20, 25, 21]. In this setting, feedback is typically provided only at the conclusion of a session, and so adapting from feedback is performed between sessions. In other words, each session consists of a non-interactive submodular optimization problem, and the technical challenge stems from the fact that the submodular function is unknown a priori and must be learned from feedback provided post optimization in each session – this setting is often referred to as inter-session interactive optimization.

The other way to make submodular optimization interactive, which we consider in this paper, is to make feedback available immediately after each action taken. In this way, one can simultaneously learn about and optimize for the unknown submodular function within a single optimization session – this setting is often referred to as intra-session interactive optimization. One can also consider settings that allow for both intra-session and inter-session interactive optimization.

Perhaps the most well-studied application of intra-session interactive submodular optimization is active learning [10, 7, 11, 9, 2, 14, 13], where the goal is to quickly reduce the hypothesis class to some target residual uncertainty for planning or decision making. Many instances of noisy and approximate active learning can be formulated as an interactive submodular set cover problem [9].

A related setting is adaptive submodularity [7, 2, 6, 13], which is a probabilistic setting that essentially requires that the conditional expectation over the hypothesis set of submodular functions is itself a submodular function. In contrast, we require that the hypothesis class be pointwise submodular (i.e., each hypothesis corresponds to a different submodular utility function). Although neither adaptive submodularity nor pointwise submodularity is a strict generalization of the other (cf. [7, 9]), in practice it can often be easier to model application settings using pointwise submodularity.

The "flipped" problem is to maximize utility with a bounded budget, which is commonly known as the budgeted submodular maximization problem [18]. Interactive budgeted maximization has been analyzed rigorously for adaptive submodular problems [7], but it remains a challenge to develop provably near-optimal interactive algorithms for pointwise submodular utility functions.

# 8 Conclusions

We introduced smooth interactive submodular set cover, a smoothed generalization of previous ISSC frameworks. Smooth ISSC allows for the target threshold to vary based on the plausibility of the hypothesis. Smooth ISSC also introduces an approximate threshold solution concept that can be applied to real-valued functions, which also applies to basic submodular set cover with real-valued objectives. We developed the first provably near-optimal algorithm for this setting.

## Footnotes

[1]When each $F_h$ and/or $G_h$ are integer-valued, then $D_F = 1$ and/or $D_G = 1$, respectively.

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
