[Supplementary Material]

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

# Supplementary Material

## A  Analysis of Multiple Thresholds Version

The following lemmas will be used to show that Definition 4.4 of $\bar{F}$ solves the equivalent problem as smooth ISSC with multiple thresholds. Furthermore, since converting smooth ISSC to $\bar{F}$ is effectively a reduction to a non-interactive submodular set cover problem, then we can prove near-optimal cost guarantees for the standard greedy algorithm.

**Lemma A.1.** *For an algorithm to ensure that $F_h(\hat{S}) \geq \alpha(G_h(S^*))$ for all $h^*$, it is both necessary and sufficient to ensure that $F_h(\hat{S}) \geq \alpha(G_h(\hat{S}))$, where $\hat{S}$ is the action set chosen by the algorithm.*

*Proof.* To show that this condition is sufficient, notice that $\hat{S} \subseteq S^*$, so $G_h(\hat{S}) \leq G_h(S^*)$. Because $\alpha(\cdot)$ is a non-increasing function, $\alpha(G_h(\hat{S})) \geq \alpha(G_h(S^*))$. Thus, if $F_h(\hat{S}) \geq \alpha(G_h(\hat{S}))$, then $F_h(\hat{S}) \geq \alpha(G_h(S^*))$.

To show that this condition is necessary, suppose there is a hypothesis $h^*$ which agrees with $h$ on all queries in $S^* \backslash \hat{S}$. For this $h^*$, $G_h(\hat{S}) = G_h(S^*)$. Thus, any hypothesis $h$ where $F_h(\hat{S}) < \alpha(G_h(\hat{S}))$ cannot be considered satisfied because there exists an $h^*$ where $F_h(\hat{S}) < \alpha(G_h(S^*))$. Thus, this condition is also necessary. $\square$

**Lemma A.2.** $\bar{F}(\hat{S}) \geq \bar{F}_{max}$ *if and only if* $F_h(\hat{S}) \geq \alpha_n$ *for all $h$ such that $G_h(S^*) < \kappa_n$ for* $n \in \{1, \dots, N\}$.

*Proof.* Due to Lemma A.1, it is equivalent to show that $\bar{F}(\hat{S}) \geq \bar{F}_{max}$ if and only if $F_h(\hat{S}) \geq \alpha_n$ for all $h$ such that $G_h(\hat{S}) < \kappa_n$ for $n \in \{1, \dots, N\}$.

First, suppose that $\bar{F}(\hat{S}) \geq \bar{F}_{max}$. $\bar{F}(\hat{S})$ may not exceed its maximum value, so

$$\bar{F}(\hat{S}) = \bar{F}_{max} = |H| D_F \alpha_1 D_G^N \prod_{n=1}^{N} (\kappa_n - \kappa_{n-1}).$$

Note that for all $h \in H$, when $\bar{F}_{h_{max}}$ is defined as the maximum value of $\bar{F}_h$,

$$0 \leq \bar{F}_h(\hat{S}) \leq \bar{F}_{h_{max}} = D_F \alpha_1 D_G^N \prod_{n=1}^{N} (\kappa_n - \kappa_{n-1}).$$

Then, if $\bar{F}(\hat{S}) = \bar{F}_{max}$, then $\bar{F}_h(\hat{S}) = \bar{F}_{h_{max}}$ for all $h \in H$.

Next, when $\bar{F}_{h,n_{max}}$ is defined as the maximum value of $\bar{F}_{h,n}$,

$$0 \leq \bar{F}_{h,n}(\hat{S}) \leq \bar{F}_{h,n_{max}} = (\alpha_n - \alpha_{n+1})(\kappa_n - \kappa_{n-1}).$$

Then, if $\bar{F}_h(\hat{S}) = \bar{F}_{h_{max}}$, then $\bar{F}_{h,n}(\hat{S}) = \bar{F}_{h,n_{max}}$ for all $h \in H$ and all $n \in \{1, 2, \dots, N\}$.

Finally, if $\bar{F}_{h,n}(\hat{S}) = \bar{F}_{h,n_{max}}$, then $F_{h,\alpha_n,\alpha_{n+1}}(\hat{S}) = \alpha_n - \alpha_{n+1}$ or $G_{h,\kappa_n,\kappa_{n-1}}(\hat{S}) = \kappa_n - \kappa_{n-1}$. If $F_{h,\alpha_n,\alpha_{n+1}}(\hat{S}) = \alpha_n - \alpha_{n+1}$, then $F_h(\hat{S}) \geq \alpha_n$, and if $G_{h,\kappa_n,\kappa_{n-1}}(\hat{S}) = \kappa_n - \kappa_{n-1}$, then $G_h(\hat{S}) \geq \kappa_n$. This implies that $F_h(\hat{S}) \geq \alpha_n$ for all $h$ such that $G_h(\hat{S}) < \kappa_n$ for $n \in \{1, \dots, N\}$.

For the opposite direction, suppose that $F_h(\hat{S}) \geq \alpha_n$ for all $h$ such that $G_h(\hat{S}) < \kappa_n$ for $n \in \{1, \dots, N\}$. This means that for all $h \in H$ and all $n \in \{1, \dots, N\}$, $F_h(\hat{S}) \geq \alpha_n$ or $G_h(\hat{S}) \geq \kappa_n$. Then, $F_{h,\alpha_n,\alpha_{n+1}}(\hat{S}) = (\alpha_n - \alpha_{n+1})$ or $G_{h,\kappa_n,\kappa_{n-1}}(\hat{S}) = (\kappa_n - \kappa_{n-1})$. Then, $\bar{F}_{h,n}(\hat{S}) = (\alpha_n - \alpha_{n+1})(\kappa_n - \kappa_{n-1})$, $\bar{F}_h(\hat{S}) = D_F \alpha_1 D_G^N \prod_{n=1}^{N} (\kappa_n - \kappa_{n-1})$, and $\bar{F}(\hat{S}) = |H| D_F \alpha_1 D_G^N \prod_{n=1}^{N} (\kappa_n - \kappa_{n-1}) = \bar{F}_{max}$. $\square$

**Lemma A.3.** *Let $F_h(\hat{S})$ and $G_h(\hat{S})$ be monotone non-decreasing submodular functions, and let the sequence $\frac{\alpha_n - \alpha_{n+1}}{\kappa_n - \kappa_{n-1}}$ for $n \in \{1, \dots, N\}$ be non-increasing [Condition 4.5]. Then, $\bar{F}(\hat{S})$ from Definition 4.4 is a monotone non-decreasing submodular function.*

*Proof.* Define $\delta_S(F, x) \triangleq F(S \oplus x) - F(S)$. First, we show that $\delta_A(\bar{F}, x) \geq 0$ for all $A$ and $x$:

$$
\begin{aligned}
\delta_A(\bar{F}, x) &= \sum_{h \in H} C_{\bar{F}} \sum_{n=1}^N \left[ \left( \prod_{j \neq n} (\kappa_j - \kappa_{j-1}) \right) \delta_A(\bar{F}_{h,n}, x) \right] \\
&= \sum_{h \in H} C_{\bar{F}} \sum_{n=1}^N \left[ \left( \prod_{j \neq n} (\kappa_j - \kappa_{j-1}) \right) \left( (\kappa_n - \kappa_{n-1}) \delta_A(F_{h,\alpha_n,\alpha_{n+1}}, x) \right. \right. \\
&\qquad\qquad\qquad\qquad\qquad\qquad + \delta_A(G_{h,\kappa_n,\kappa_{n-1}}, x)(\alpha_n - \alpha_{n+1}) \\
&\qquad\qquad\qquad\qquad\qquad\qquad + F_{h,\alpha_n,\alpha_{n+1}}(A) G_{h,\kappa_n,\kappa_{n-1}}(A) \\
&\qquad\qquad\qquad\qquad\qquad\qquad \left. \left. - F_{h,\alpha_n,\alpha_{n+1}}(A \oplus x) G_{h,\kappa_n,\kappa_{n-1}}(A \oplus x) \right) \right] \\
&= \sum_{h \in H} C_{\bar{F}} \sum_{n=1}^N \left[ \left( \prod_{j \neq n} (\kappa_j - \kappa_{j-1}) \right) \left( ((\kappa_n - \kappa_{n-1}) - G_{h,\kappa_n,\kappa_{n-1}}(A)) \delta_A(F_{h,\alpha_n,\alpha_{n+1}}, x) \right. \right. \\
&\qquad\qquad\qquad\qquad\qquad\qquad \left. \left. + \delta_A(G_{h,\kappa_n,\kappa_{n-1}}, x)((\alpha_n - \alpha_{n+1}) - F_{h,\alpha_n,\alpha_{n+1}}(A \oplus x)) \right) \right]
\end{aligned}
$$

Note that $(\kappa_n - \kappa_{n-1}) - G_{h,\kappa_n,\kappa_{n-1}}(A)$, $\delta_A(F_{h,\alpha_n,\alpha_{n+1}}, x)$, $\delta_A(G_{h,\kappa_n,\kappa_{n-1}}, x)$, and $(\alpha_n - \alpha_{n+1}) - F_{h,\alpha_n,\alpha_{n+1}}(A \oplus x)$ are all non-negative. Thus, $\delta_A(\bar{F}_h, x) \geq 0$, and $\bar{F}_h(S)$ is non-decreasing.

Next, consider any $B$ such that $A \subseteq B$. Similarly,

$$
\begin{aligned}
\delta_B(\bar{F}, x) &\triangleq \sum_{h \in H} C_{\bar{F}} \sum_{n=1}^N \left[ \left( \prod_{j \neq n} (\kappa_j - \kappa_{j-1}) \right) \left( ((\kappa_n - \kappa_{n-1}) - G_{h,\kappa_n,\kappa_{n-1}}(B)) \delta_B(F_{h,\alpha_n,\alpha_{n+1}}, x) \right. \right. \\
&\qquad\qquad\qquad\qquad\qquad\qquad \left. \left. + \delta_B(G_{h,\kappa_n,\kappa_{n-1}}, x)((\alpha_n - \alpha_{n+1}) - F_{h,\alpha_n,\alpha_{n+1}}(B \oplus x)) \right) \right]
\end{aligned}
$$

Then, we show that for all $A \subseteq B$, $\delta_B(\bar{F}, x) - \delta_A(\bar{F}, x) \geq 0$.

$$
\begin{aligned}
\delta_B(\bar{F}, x) - \delta_A(\bar{F}, x) &\triangleq \sum_{h \in H} C_{\bar{F}} \sum_{n=1}^N \left[ \left( \prod_{j \neq n} (\kappa_j - \kappa_{j-1}) \right) \right. \\
&\qquad \left( (\kappa_n - \kappa_{n-1})(\delta_B(F_{h,\alpha_n,\alpha_{n+1}}, x) - \delta_A(F_{h,\alpha_n,\alpha_{n+1}}, x)) \right. \\
&\qquad\quad - G_{h,\kappa_n,\kappa_{n-1}}(B) \delta_B(F_{h,\alpha_n,\alpha_{n+1}}, x) \\
&\qquad\quad + G_{h,\kappa_n,\kappa_{n-1}}(A) \delta_A(F_{h,\alpha_n,\alpha_{n+1}}, x) \\
&\qquad\quad + (\alpha_n - \alpha_{n+1})(\delta_B(G_{h,\kappa_n,\kappa_{n-1}}, x) - \delta_A(G_{h,\kappa_n,\kappa_{n-1}}, x)) \\
&\qquad\quad - \delta_B(G_{h,\kappa_n,\kappa_{n-1}}, x) F_{h,\alpha_n,\alpha_{n+1}}(B \oplus x) \\
&\qquad\quad \left. \left. + \delta_A(G_{h,\kappa_n,\kappa_{n-1}}, x) F_{h,\alpha_n,\alpha_{n+1}}(A \oplus x) \right) \right]
\end{aligned}
$$

Note that $G_{h,\kappa_n,\kappa_{n+1}}(A) \leq G_{h,\kappa_n,\kappa_{n+1}}(B)$ and $F_{h,\alpha_n,\alpha_{n+1}}(A \oplus x) \leq F_{h,\alpha_n,\alpha_{n+1}}(B \oplus x)$. Then, $\delta_B(\bar{F}, x) - \delta_A(\bar{F}, x) \leq \sum_{h \in H} C_{\bar{F}} \mathbb{L}_h$, where,

$$
\begin{aligned}
\mathbb{L}_h &= \sum_{n=1}^N \left[ \left( \prod_{j \neq n} (\kappa_j - \kappa_{j-1}) \right) \right. \\
&\qquad \left( (\kappa_n - \kappa_{n-1})(\delta_B(F_{h,\alpha_n,\alpha_{n+1}}, x) - \delta_A(F_{h,\alpha_n,\alpha_{n+1}}, x)) \right. \\
&\qquad\quad - G_{h,\kappa_n,\kappa_{n-1}}(B) \delta_B(F_{h,\alpha_n,\alpha_{n+1}}, x) + G_{h,\kappa_n,\kappa_{n-1}}(B) \delta_A(F_{h,\alpha_n,\alpha_{n+1}}, x) \\
&\qquad\quad + (\alpha_n - \alpha_{n+1})(\delta_B(G_{h,\kappa_n,\kappa_{n-1}}, x) - \delta_A(G_{h,\kappa_n,\kappa_{n-1}}, x)) \\
&\qquad\quad \left. \left. - \delta_B(G_{h,\kappa_n,\kappa_{n-1}}, x) F_{h,\alpha_n,\alpha_{n+1}}(B \oplus x) + \delta_A(G_{h,\kappa_n,\kappa_{n-1}}, x) F_{h,\alpha_n,\alpha_{n+1}}(B \oplus x) \right) \right]
\end{aligned}
$$

$$= \sum_{n=1}^{N} \left[ \left( \prod_{j \neq n} (\kappa_j - \kappa_{j-1}) \right) \right.$$

$$\left( ((\kappa_n - \kappa_{n-1}) - G_{h,\kappa_n,\kappa_{n-1}}(B))(\delta_B(F_{h,\alpha_n,\alpha_{n+1}}, x) - \delta_A(F_{h,\alpha_n,\alpha_{n+1}}, x)) \right.$$

$$\left. \left. + ((\alpha_n - \alpha_{n+1}) - F_{h,\alpha_n,\alpha_{n+1}}(B \oplus x))(\delta_B(G_{h,\kappa_n,\kappa_{n-1}}, x) - \delta_A(G_{h,\kappa_n,\kappa_{n-1}}, x)) \right) \right]$$

$$= \left( \prod_{j=1}^{N} (\kappa_j - \kappa_{j-1}) \right)$$

$$\left[ \sum_{n=1}^{N} \frac{(\kappa_n - \kappa_{n-1}) - G_{h,\kappa_n,\kappa_{n-1}}(B)}{\kappa_n - \kappa_{n-1}} (\delta_B(F_{h,\alpha_n,\alpha_{n+1}}, x) - \delta_A(F_{h,\alpha_n,\alpha_{n+1}}, x)) \right.$$

$$\left. + \sum_{n=1}^{N} \frac{(\alpha_n - \alpha_{n+1}) - F_{h,\alpha_n,\alpha_{n+1}}(B \oplus x)}{\kappa_n - \kappa_{n-1}} (\delta_B(G_{h,\kappa_n,\kappa_{n-1}}, x) - \delta_A(G_{h,\kappa_n,\kappa_{n-1}}, x)) \right].$$

Note that the sequence $\frac{(\kappa_n - \kappa_{n-1}) - G_{h,\kappa_n,\kappa_{n-1}}(B)}{\kappa_n - \kappa_{n-1}}$ must take the form $\langle 0, \dots, 0, a, 1, \dots, 1 \rangle$ where $a \in [0, 1]$. In addition, $\sum_{n=1}^{N} \delta_A(F_{h,\alpha_n,\alpha_{n+1}}, x) \geq \sum_{n=1}^{j} \delta_B(F_{h,\alpha_n,\alpha_{n+1}}, x)$ for all positive integer $j \leq N$. Thus, $\sum_{n=1}^{N} \frac{(\kappa_n - \kappa_{n-1}) - G_{h,\kappa_n,\kappa_{n-1}}(B)}{\kappa_n - \kappa_{n-1}} (\delta_B(F_{h,\alpha_n,\alpha_{n+1}}, x) - \delta_A(F_{h,\alpha_n,\alpha_{n+1}}, x))$ is non-positive.

Note also that the sequence $\frac{(\alpha_n - \alpha_{n+1}) - F_{h,\alpha_n,\alpha_{n+1}}(B \oplus x)}{\kappa_n - \kappa_{n-1}}$ must take the form $\langle \frac{\alpha_1 - \alpha_2}{\kappa_1 - \kappa_0}, \dots, \frac{\alpha_{n-1} - \alpha_n}{\kappa_{n-1} - \kappa_{n-2}}, a, 0, \dots, 0 \rangle$ where $a \in [0, \frac{\alpha_n - \alpha_{n+1}}{\kappa_n - \kappa_{n-1}}]$. Because of the restriction on the values of $\frac{\alpha_n - \alpha_{n+1}}{\kappa_n - \kappa_{n-1}}$ (Condition 4.5), this sequence is non-increasing. In addition, $\sum_{n=1}^{j} \delta_A(G_{h,\kappa_n,\kappa_{n-1}}, x) \geq \sum_{n=1}^{j} \delta_B(G_{h,\kappa_n,\kappa_{n-1}}, x)$ for all positive integer $j \leq N$. Thus, $\sum_{n=1}^{N} \frac{(\alpha_n - \alpha_{n+1}) - F_{h,\alpha_n,\alpha_{n+1}}(B \oplus x)}{\kappa_n - \kappa_{n-1}} (\delta_B(G_{h,\kappa_n,\kappa_{n-1}}, x) - \delta_A(G_{h,\kappa_n,\kappa_{n-1}}, x))$ is non-positive.

These two statements imply that $\delta_B(F, x) - \delta_A(F, x) \leq 0$, which means that $F$ is submodular. $\square$

**Lemma A.4** (Lemma 3 from [10]). *For any initial set of questions-response pairs $\hat{S}$, there must be a question $q \in Q$ such that*

$$\min_{r \in \mathcal{R}} \left( \bar{F}(\hat{S} \oplus (q, r)) - \bar{F}(\hat{S}) \right) \geq c(q)(\bar{F}_{max} - \bar{F}(\hat{S}))/GCC$$

*Proof.* Assume that the lemma is false and for every question $q$, there is some $r \in \mathcal{R}$ such that

$$\bar{F}(\hat{S} \oplus (q, r)) - \bar{F}(\hat{S}) < c(q)(\bar{F}_{max} - \bar{F}(\hat{S}))/GCC$$

Define an oracle $T'$ which answers every question with a response satisfying this inequality. For example, one such $T'$ is

$$T'(q) \triangleq \operatorname{argmin}_r \left( \bar{F}(\hat{S} \oplus (q, r)) - \bar{F}(\hat{S}) \right)$$

By the definition of $GCC$,

$$\min_{\hat{Q}: \bar{F}(T'(\hat{Q})) \geq \bar{F}_{max}} c(\hat{Q}) \leq \max_{T \in \mathcal{R}^Q} \left( \min_{\hat{Q}: \bar{F}(T(\hat{Q})) \geq \bar{F}_{max}} c(\hat{Q}) \right) = GCC$$

so there must be a sequence of questions $\hat{Q}$ with $c(\hat{Q}) \leq GCC$ such that $\bar{F}(T'(\hat{Q})) \geq \bar{F}_{max}$. Because $\bar{F}$ is monotone non-decreasing, we also know that $\bar{F}(T'(\hat{Q}) \cup \hat{S}) \geq \bar{F}_{max}$. Using the submodularity of $\bar{F}$,

$$\bar{F}(T'(\hat{Q}) \cup \hat{S}) \leq \bar{F}(\hat{S}) + \sum_{q \in \hat{Q}} (\bar{F}(\hat{S} \cup \{(q, T(q))\}) - \bar{F}(\hat{S}))$$

$$< \bar{F}(\hat{S}) + \sum_{q \in \hat{Q}} c(q)(\bar{F}_{max} - \bar{F}(\hat{S}))/GCC \leq \bar{F}_{max}$$

which is a contradiction. $\qquad\square$

**Theorem A.5** (Theorem 1 from [10]). *Assume that $\bar{F}$ is an integral monotone non-decreasing submodular function. Algorithm 1 incurs at most $GCC(1 + \ln(\bar{F}_{max}))$ cost.*

*Proof.* Let $\hat{q}_i$ be the question asked on the $i$th iteration, $\hat{S}_i$ be the set of question-response pairs after asking $\hat{q}_i$ and $C_i$ be $\sum_{j \leq i} c(\hat{q}_j)$. By Lemma A.4,

$$\bar{F}(\hat{S}_i) - \bar{F}(\hat{S}_{i-1}) \geq c(\hat{q}_i)(\bar{F}_{max} - \bar{F}(\hat{S}_{i-1}))/GCC$$

After some algebra we get

$$\bar{F}_{max} - \bar{F}(\hat{S}_i) \leq (\bar{F}_{max} - \bar{F}(\hat{S}_{i-1}))(1 - c(\hat{q}_i)/GCC)$$

Now using $1 - x < e^{-x}$

$$\bar{F}_{max} - \bar{F}(\hat{S}_i) \leq (\bar{F}_{max} - \bar{F}(\hat{S}_{i-1}))e^{-c(\hat{q}_i)/GCC} \leq \bar{F}_{max}e^{-C_i/GCC}$$

We have shown that the gap $\bar{F}_{max} - \bar{F}(\hat{S}_i)$ decreases exponentially fast with the cost of the questions asked. The remainder of the proof proceeds by showing that (1) we can decrease the gap to 1 using questions with at most $GCC \ln(\bar{F}_{max})$ cost and (2) we can decrease the gap from 1 to 0 with one question with cost at most $GCC$.

Let $j$ be the largest integer such that $\bar{F}_{max} - \bar{F}(\hat{S}_j) \geq 1$ holds. Then

$$1 \leq \bar{F}_{max}e^{-C_j/GCC}$$

Solving for $C_j$ we get $C_j \leq GCC \ln(\bar{F}_{max})$. This completes (1).

By Lemma A.4, $\bar{F}(\hat{S}_i) < \bar{F}(\hat{S}_{i+1})$ (we strictly increase the objective on each iteration). Because $\bar{F}_{max}$ is an integer and $\bar{F}$ is an integral function, we can conclude that $\bar{F}(\hat{S}_i) \leq \bar{F}(\hat{S}_{i+1} + 1)$. Then $q_{j+1}$ will be the final question asked. By Lemma A.4, $q_{j+1}$ can have cost no greater than $GCC$. This completes (2). We can finally conclude the cost incurred by the greedy algorithm is at most $GCC(1 + \ln(\bar{F}_{max}))$. $\qquad\square$

*Proof of Theorem 4.6.* Lemma A.2 implies that satisfying the condition $\bar{F} \geq \bar{F}_{max}$ is equivalent to satisfying the goal of smooth ISSC with multiple thresholds. Next, Lemma A.3 implies that $\bar{F}$ may be used with Algorithm 1 and have guaranteed performance bounds. Finally, Theorem A.5 shows that the upper bound of Algorithm 1 is $\left(1 + \ln(\bar{F}_{max})\right)GCC$. Plugging in the value of $\bar{F}_{max}$, the upper bound in the multiple threshold case of smooth ISSC is then:

$$\left(1 + \ln(\bar{F}_{max})\right)GCC = \left(1 + \ln\left(|H|D_F D_G^N \alpha_1 \prod_{n=1}^{N}(\kappa_n - \kappa_{n-1})\right)\right)GCC,$$

giving Theorem 4.6. $\qquad\square$

# B  Analysis of Approximate Thresholds for Real-Valued Functions

In this section, we show that the surrogate problem in Definition 4.7 is an instance of a smooth ISSC problem, in particular one with approximate thresholds compared to the original problem.

**Lemma B.1.** $F'_h(\hat{S}) \geq \alpha'_n$ *implies that* $F_h(\hat{S}) \geq \alpha_n - \epsilon$.

*Proof.* Suppose that $F'_h(\hat{S}) \geq \alpha'_n$. Then,

$$F'_h(\hat{S}) \geq \alpha'_n,$$

$$\frac{D}{\epsilon}\left\lfloor F_h(\hat{S}) + \frac{\epsilon}{D}\sum_{i=1}^{|\hat{S}|}(|\mathcal{Q}| + 1 - i)\right\rfloor_{\frac{\epsilon}{D}} \geq \frac{D}{\epsilon}\left\lfloor \alpha_n - \frac{\epsilon}{D}\sum_{i=1}^{n}\left[(2N - 2i)D_G^{N-i+1}\prod_{j=i}^{N}(\kappa_j - \kappa_{j-1})\right]\right\rfloor_{\frac{\epsilon}{D}},$$

$$\left\lceil F_h(\hat{S}) + \frac{\epsilon}{D}\sum_{i=1}^{|\hat{S}|}(|\mathcal{Q}|+1-i)\right\rceil_{\frac{\epsilon}{D}} \geq \left\lfloor \alpha_n - \frac{\epsilon}{D}\sum_{i=1}^{n}\left[(2N-2i)D_G^{N-i+1}\prod_{j=i}^{N}(\kappa_j-\kappa_{j-1})\right]\right\rfloor_{\frac{\epsilon}{D}},$$

$$F_h(\hat{S}) + \frac{\epsilon}{D}\sum_{i=1}^{|\hat{S}|}(|\mathcal{Q}|+1-i) + \frac{\epsilon}{D} \geq \left\lfloor \alpha_n - \frac{\epsilon}{D}\sum_{i=1}^{n}\left[(2N-2i)D_G^{N-i+1}\prod_{j=i}^{N}(\kappa_j-\kappa_{j-1})\right]\right\rfloor_{\frac{\epsilon}{D}},$$

$$F_h(\hat{S}) + \frac{\epsilon}{D}\sum_{i=1}^{|\hat{S}|}(|\mathcal{Q}|+1-i) + \frac{\epsilon}{D} \geq \alpha_n - \frac{\epsilon}{D}\sum_{i=1}^{n}\left[(2N-2i)D_G^{N-i+1}\prod_{j=i}^{N}(\kappa_j-\kappa_{j-1})\right] - \frac{\epsilon}{D},$$

$$F_h(\hat{S}) \geq \alpha_n - \frac{\epsilon}{D}\left[\sum_{i=1}^{|\hat{S}|}(|\mathcal{Q}|+1-i) + \sum_{i=1}^{n}\left[(2N-2i)D_G^{N-i+1}\prod_{j=i}^{N}(\kappa_j-\kappa_{j-1})\right] + 2\right],$$

$$F_h(\hat{S}) \geq \alpha_n - \epsilon.$$

$\square$

**Lemma B.2.** $F_h'(\hat{S}) < \alpha_i'$ *implies that* $F_h(\hat{S}) < \alpha_i$.

*Proof.* Suppose that $F_h'(\hat{S}) < \alpha_n'$. Then,

$$F_h'(\hat{S}) < \alpha_n',$$

$$\frac{D}{\epsilon}\left\lceil F_h(\hat{S}) + \frac{\epsilon}{D}\sum_{i=1}^{|\hat{S}|}(|\mathcal{Q}|+1-i)\right\rceil_{\frac{\epsilon}{D}} < \frac{D}{\epsilon}\left\lfloor \alpha_n - \frac{\epsilon}{D}\sum_{i=1}^{n}\left[(2N-2i)D_G^{N-i+1}\prod_{j=i}^{N}(\kappa_j-\kappa_{j-1})\right]\right\rfloor_{\frac{\epsilon}{D}},$$

$$\left\lceil F_h(\hat{S}) + \frac{\epsilon}{D}\sum_{i=1}^{|\hat{S}|}(|\mathcal{Q}|+1-i)\right\rceil_{\frac{\epsilon}{D}} < \left\lfloor \alpha_n - \frac{\epsilon}{D}\sum_{i=1}^{n}\left[(2N-2i)D_G^{N-i+1}\prod_{j=i}^{N}(\kappa_j-\kappa_{j-1})\right]\right\rfloor_{\frac{\epsilon}{D}},$$

$$F_h(\hat{S}) + \frac{\epsilon}{D}\sum_{i=1}^{|\hat{S}|}(|\mathcal{Q}|+1-i) < \alpha_n - \frac{\epsilon}{D}\sum_{i=1}^{n}\left[(2N-2i)D_G^{N-i+1}\prod_{j=i}^{N}(\kappa_j-\kappa_{j-1})\right],$$

$$F_h(\hat{S}) < \alpha_n - \frac{\epsilon}{D}\sum_{i=1}^{n}\left[(2N-2i)D_G^{N-i+1}\prod_{j=i}^{N}(\kappa_j-\kappa_{j-1})\right],$$

$$F_h(\hat{S}) < \alpha_n.$$

$\square$

**Lemma B.3.** $F_h'$ *preserves monotonicity and submodularity of* $F_h$.

*Proof.* Define $\delta_S(F,x) = F(S \oplus x) - F(S)$.

First, assume that $F_h$ is monotone non-decreasing, which implies that $\delta_S(F_h, x) \geq 0$. Then,

$$\delta_S(F_h', x) = \frac{D}{\epsilon}\left\lceil F_h(S \oplus x) + \frac{\epsilon}{D}\sum_{i=1}^{|S|+\{x\}}(|\mathcal{Q}|+1-i)\right\rceil_{\frac{\epsilon}{D}} - \frac{D}{\epsilon}\left\lceil F_h(S) + \frac{\epsilon}{D}\sum_{i=1}^{|S|}(|\mathcal{Q}|+1-i)\right\rceil_{\frac{\epsilon}{D}}$$

$$= \frac{D}{\epsilon}\left[\left\lceil F_h(S \oplus x) + \frac{\epsilon}{D}\sum_{i=1}^{|S|+\{x\}}(|\mathcal{Q}|+1-i)\right\rceil_{\frac{\epsilon}{D}} - \left\lceil F_h(S) + \frac{\epsilon}{D}\sum_{i=1}^{|S|}(|\mathcal{Q}|+1-i)\right\rceil_{\frac{\epsilon}{D}}\right]$$

$$= \frac{D}{\epsilon}\left[\lceil F_h(S \oplus x)\rceil_{\frac{\epsilon}{D}} + \frac{\epsilon}{D}\sum_{i=1}^{|S|+\{x\}}(|\mathcal{Q}|+1-i) - \lceil F_h(S)\rceil_{\frac{\epsilon}{D}} - \frac{\epsilon}{D}\sum_{i=1}^{|S|}(|\mathcal{Q}|+1-i)\right]$$

$$= \frac{D}{\epsilon}\left[\lceil F_h(S \oplus x)\rceil_{\frac{\epsilon}{D}} - \lceil F_h(S)\rceil_{\frac{\epsilon}{D}} + \frac{\epsilon}{D}\sum_{i=1}^{|S|+\{x\}}(|\mathcal{Q}|+1-i) - \frac{\epsilon}{D}\sum_{i=1}^{|S|}(|\mathcal{Q}|+1-i)\right]$$

$$= \frac{D}{\epsilon}\left[\lceil F_h(S \oplus x)\rceil_{\frac{\epsilon}{D}} - \lceil F_h(S)\rceil_{\frac{\epsilon}{D}} + \frac{\epsilon}{D}(|\mathcal{Q}|-|S|)\right],$$

because $\delta_S(F_h, x) \geq 0$, $F_h(S \oplus x) \geq F_h(S)$, and $\lceil F_h(S \oplus x) \rceil \geq \lceil F_h(S) \rceil$. In addition, $|\mathcal{Q}| \geq |S|$. Thus, $\delta_S(F'_h, x) \geq 0$, and $F'_h$ is non-decreasing.

Next, $F_h$ being submodular implies that for $A \subseteq B$, $\delta_A(F_h, x) = F_h(A + \{x\}) - F_h(A) \geq F_h(B + \{x\}) - F_h(B) = \delta_B(F_h, x)$. Then,

$$\lceil F_h(A + \{x\}) \rceil_{\frac{\epsilon}{D}} - \lceil F_h(A) \rceil_{\frac{\epsilon}{D}} + \frac{\epsilon}{D} \geq \lceil F_h(B + \{x\}) \rceil_{\frac{\epsilon}{D}} - \lceil F_h(B) \rceil_{\frac{\epsilon}{D}},$$

$$\lceil F_h(A + \{x\}) \rceil_{\frac{\epsilon}{D}} - \lceil F_h(A) \rceil_{\frac{\epsilon}{D}} + \frac{\epsilon}{D}(|\mathcal{Q}| - |A|) \geq \lceil F_h(B + \{x\}) \rceil_{\frac{\epsilon}{D}} - \lceil F_h(B) \rceil_{\frac{\epsilon}{D}} + \frac{\epsilon}{D}(|\mathcal{Q}| - |B|),$$

$$\frac{D}{\epsilon}\left[ \lceil F_h(A + \{x\}) \rceil_{\frac{\epsilon}{D}} - \lceil F_h(A) \rceil_{\frac{\epsilon}{D}} + \frac{\epsilon}{D}(|\mathcal{Q}| - |A|) \right] \geq \frac{D}{\epsilon}\left[ \lceil F_h(B + \{x\}) \rceil_{\frac{\epsilon}{D}} - \lceil F_h(B) \rceil_{\frac{\epsilon}{D}} + \frac{\epsilon}{D}(|\mathcal{Q}| - |B|) \right],$$

$$\delta_A(F'_h, x) \geq \delta_B(F'_h, x).$$

$\square$

**Lemma B.4.** *If $\langle \alpha_n \rangle_{n=1}^N$ is decreasing, then $\langle \alpha'_n \rangle_{n=1}^N$ is decreasing.*

*Proof.*

$$\alpha'_n \triangleq \frac{D}{\epsilon}\left\lfloor \alpha_n - \frac{\epsilon}{D}\sum_{i=1}^n \left[ (2N - 2i)D_G^{N-i+1}\prod_{j=i}^N(\kappa_j - \kappa_{j-1}) \right] \right\rfloor_{\frac{\epsilon}{D}}$$

$$\geq \frac{D}{\epsilon}\left\lfloor \alpha_{n+1} - \frac{\epsilon}{D}\sum_{i=1}^n \left[ (2N - 2i)D_G^{N-i+1}\prod_{j=i}^N(\kappa_j - \kappa_{j-1}) \right] \right\rfloor_{\frac{\epsilon}{D}}$$

$$> \frac{D}{\epsilon}\left\lfloor \alpha_{n+1} - \frac{\epsilon}{D}\sum_{i=1}^{n+1} \left[ (2N - 2i)D_G^{N-i+1}\prod_{j=i}^N(\kappa_j - \kappa_{j-1}) \right] \right\rfloor_{\frac{\epsilon}{D}}$$

$$= \alpha'_{n+1}$$

$\square$

**Lemma B.5.** *If $\langle \frac{\alpha_n - \alpha_{n+1}}{\kappa_n - \kappa_{n-1}} \rangle_{n=1}^N$ is non-increasing, then $\langle \frac{\alpha'_n - \alpha'_{n+1}}{\kappa_n - \kappa_{n-1}} \rangle_{n=1}^N$ is also non-increasing.*

*Proof.*

$$\alpha'_n - \alpha'_{n+1} = \frac{D}{\epsilon}\left\lfloor \alpha_n - \frac{\epsilon}{D}\sum_{i=1}^n \left[ (2N - 2i)D_G^{N-i+1}\prod_{j=i}^N(\kappa_j - \kappa_{j-1}) \right] \right\rfloor_{\frac{\epsilon}{D}}$$

$$- \frac{D}{\epsilon}\left\lfloor \alpha_{n+1} - \frac{\epsilon}{D}\sum_{i=1}^{n+1} \left[ (2N - 2i)D_G^{N-i+1}\prod_{j=i}^N(\kappa_j - \kappa_{j-1}) \right] \right\rfloor_{\frac{\epsilon}{D}}$$

$$= \frac{D}{\epsilon}\left[ \left\lfloor \alpha_n - \frac{\epsilon}{D}\sum_{i=1}^n \left[ (2N - 2i)D_G^{N-i+1}\prod_{j=i}^N(\kappa_j - \kappa_{j-1}) \right] \right\rfloor_{\frac{\epsilon}{D}} \right.$$

$$\left. - \left\lfloor \alpha_{n+1} - \frac{\epsilon}{D}\sum_{i=1}^{n+1} \left[ (2N - 2i)D_G^{N-i+1}\prod_{j=i}^N(\kappa_j - \kappa_{j-1}) \right] \right\rfloor_{\frac{\epsilon}{D}} \right]$$

$$= \frac{D}{\epsilon}\left[ \lfloor \alpha_n \rfloor_{\frac{\epsilon}{D}} - \lfloor \alpha_{n+1} \rfloor_{\frac{\epsilon}{D}} \right.$$

$$- \frac{\epsilon}{D}\sum_{i=1}^n \left[ (2N - 2i)D_G^{N-i+1}\prod_{j=i}^N(\kappa_j - \kappa_{j-1}) \right]$$

$$\left. + \frac{\epsilon}{D}\sum_{i=1}^{n+1} \left[ (2N - 2i)D_G^{N-i+1}\prod_{j=i}^N(\kappa_j - \kappa_{j-1}) \right] \right]$$

$$= \frac{D}{\epsilon}\left[ \lfloor \alpha_n \rfloor_{\frac{\epsilon}{D}} - \lfloor \alpha_{n+1} \rfloor_{\frac{\epsilon}{D}} + \frac{\epsilon}{D}\left[ (2N - 2(n+1))D_G^{N-(n+1)+1}\prod_{j=n+1}^N(\kappa_j - \kappa_{j-1}) \right] \right].$$

Similarly,

$$\alpha'_{n+1} - \alpha'_{n+2} = \frac{D}{\epsilon}\left[\lfloor\alpha_{n+1}\rfloor_{\frac{\epsilon}{D}} - \lfloor\alpha_{n+2}\rfloor_{\frac{\epsilon}{D}} + \frac{\epsilon}{D}\left[(2N - 2(n+2))D_G^{N-(n+2)+1}\prod_{j=n+2}^{N}(\kappa_j - \kappa_{j-1})\right]\right].$$

Then,

$$
\begin{aligned}
\frac{\alpha'_n - \alpha'_{n+1}}{\kappa_n - \kappa_{n-1}} &= \frac{\frac{D}{\epsilon}\left[\lfloor\alpha_n\rfloor_{\frac{\epsilon}{D}} - \lfloor\alpha_{n+1}\rfloor_{\frac{\epsilon}{D}} + \frac{\epsilon}{D}\left[(2N - 2(n+1))D_G^{N-(n+1)+1}\prod_{j=n+1}^{N}(\kappa_j - \kappa_{j-1})\right]\right]}{\kappa_n - \kappa_{n-1}} \\[2mm]
&\geq \frac{\frac{D}{\epsilon}\left[\alpha_n - \alpha_{n+1} - \frac{\epsilon}{D} + \frac{\epsilon}{D}\left[(2N - 2(n+1))D_G^{N-(n+1)+1}\prod_{j=n+1}^{N}(\kappa_j - \kappa_{j-1})\right]\right]}{\kappa_n - \kappa_{n-1}} \\[2mm]
&\geq \frac{\frac{D}{\epsilon}\left[\alpha_n - \alpha_{n+1} + \frac{\epsilon}{D}\left[(2N - 2(n+1) - 1)D_G^{N-(n+1)+1}\prod_{j=n+1}^{N}(\kappa_j - \kappa_{j-1})\right]\right]}{\kappa_n - \kappa_{n-1}} \\[2mm]
&\geq \frac{\frac{D}{\epsilon}\left[\alpha_{n+1} - \alpha_{n+2} + \frac{\epsilon}{D}\left[(2N - 2(n+1) - 1)D_G^{N-(n+1)+1}(\kappa_{n+1} - \kappa_n)\prod_{j=n+2}^{N}(\kappa_j - \kappa_{j-1})\right]\right]}{\kappa_{n+1} - \kappa_n} \\[2mm]
&\geq \frac{\frac{D}{\epsilon}\left[\alpha_{n+1} - \alpha_{n+2} + \frac{\epsilon}{D}\left[(2N - 2(n+1) - 1)D_G^{N-(n+2)+1}\prod_{j=n+2}^{N}(\kappa_j - \kappa_{j-1})\right]\right]}{\kappa_{n+1} - \kappa_n} \\[2mm]
&\geq \frac{\frac{D}{\epsilon}\left[\lfloor\alpha_{n+1}\rfloor_{\frac{\epsilon}{D}} - \lfloor\alpha_{n+2}\rfloor_{\frac{\epsilon}{D}} - \frac{\epsilon}{D} + \frac{\epsilon}{D}\left[(2N - 2(n+1) - 1)D_G^{N-(n+2)+1}\prod_{j=n+2}^{N}(\kappa_j - \kappa_{j-1})\right]\right]}{\kappa_{n+1} - \kappa_n} \\[2mm]
&\geq \frac{\frac{D}{\epsilon}\left[\lfloor\alpha_{n+1}\rfloor_{\frac{\epsilon}{D}} - \lfloor\alpha_{n+2}\rfloor_{\frac{\epsilon}{D}} + \frac{\epsilon}{D}\left[(2N - 2(n+2))D_G^{N-(n+2)+1}\prod_{j=n+2}^{N}(\kappa_j - \kappa_{j-1})\right]\right]}{\kappa_{n+1} - \kappa_n} \\[2mm]
&= \frac{\alpha'_{n+1} - \alpha'_{n+2}}{\kappa_{n+1} - \kappa_n}
\end{aligned}
$$

$\square$

As such, solving the surrogate smooth ISSC problem in Definition 4.7 will lead to approximately solving the original real-valued smooth ISSC problem, provided Condition 4.5 holds.

*Proof of Theorem 4.8.* Lemma B.1 guarantees that solving the surrogate problem also solves the original problem to within $\epsilon$ tolerance. Lemma B.2 guarantees that the surrogate problem will be solved as long as the original problem is solved. Lemmas B.3, B.4, and B.5 show that if the original problem results in a submodular $\bar{F}$, then the surrogate problem also results in a submodular $\bar{F}$.

As discussed in the proof of Theorem 4.6, the upper bound of Algorithm 1 derived in [10] is $\left(1 + \ln(\bar{F}_{max})\right)GCC$. Notice that Lemma B.2 means that the surrogate problem is no harder to solve than the original problem. The upper bound in the approximate thresholds case is then no larger than

$$
\begin{aligned}
\left(1 + \ln(\bar{F}_{max})\right)GCC &= (1 + \ln(|H|C_F C_G))\,GCC \\
&= \left(1 + \ln\left(|H|\alpha'_1 D_G^N \alpha_1 \prod_{n=1}^{N}(\kappa_n - \kappa_{n-1})\right)\right)GCC,
\end{aligned}
$$

giving Theorem 4.8. $\square$

## B.1 Pathological Case for Real-Valued Submodular Set Cover

There are $N + 3$ items with uniform cost which are denoted $x_1, \ldots, x_N, y_1, y_2, y_3$. Define $F(S) = 1 - \left(1 - \frac{|y|}{3}\right) \left(\frac{1}{2}\right)^{|x|}$, where $|x|$ is the number of $x$ terms in $S$, and $|y|$ is the number of $y$ terms in $S$. First, adding an extra element will only increase the value of $F$, so $F$ is increasing. Next, consider adding an extra $x$. The original value is $1 - \left(1 - \frac{|y|}{3}\right) \left(\frac{1}{2}\right)^{|x|}$ and the new value is $1 - \left(1 - \frac{|y|}{3}\right) \left(\frac{1}{2}\right)^{|x|+1}$. The difference is $\left(1 - \frac{|y|}{3}\right) \left(\frac{1}{2}\right)^{|x|+1}$. Note that this is always positive and decreases when there are more elements originally in the set. Finally, consider adding an extra $y$. The new value is $1 - \left(1 - \frac{|y|+1}{3}\right) \left(\frac{1}{2}\right)^{|x|}$. The difference is $\frac{1}{3} \left(\frac{1}{2}\right)^{|x|}$. Again, this is always positive and decreases when there are more elements originally in the set. Thus, $F$ is a monotone non-decreasing submodular function.

Next, notice the smallest set $S$ such that $F(S) \geq 1$ is $S = \{y_1, y_2, y_3\}$. However, the greedy algorithm will take all of the $x_i$ before taking any of the $y_i$ due to the fact that taking a $x_i$ term will increase $F$ by $\frac{1}{2}$ of the remaining value, whereas taking a $y_i$ term will increase $F$ by only $\frac{1}{3}$ of the remaining value.

## B.2 Reduction to Basic Submodular Set Cover with Real-Valued Objective Functions

In the basic submodular set cover problem, we have a single submodular function $F$ that we need to satisfy a threshold $\alpha$ with minimum cost. To apply the solution to the approximate multiple thresholds version of smooth ISSC to basic submodular set cover, the following definition is used.

**Definition B.6** (Reduction to Basic Submodular Set Cover)**.** *Definition 4.7 is instantiated with the following choices of parameters to solve the equivalent basic submodular set cover problem.*

$$|H| = 1, \quad N = 1,$$
$$F_h(\hat{S}) = F(\hat{S}), \quad G_h(\hat{S}) = 0,$$
$$\alpha_1 = \alpha, \quad \kappa_1 = 1, \quad D_G = 1$$

**Lemma B.7.** *Solving the basic submodular set cover problem is equivalent to solving the instantiation of smooth ISSC in Defnition B.6.*

*Proof.* If the basic problem is solved, then $F(\hat{S}) \geq \alpha$. This implies that $F_h(\hat{S}) \geq \alpha_1$, so the smooth ISSC problem is also solved.

Next, if the smooth ISSC problem is solved, then $F_h(\hat{S}) \geq \alpha_1$ or $G_h(\hat{S}) \geq \kappa_1$. However, $G_h(\hat{S}) = 0 < 1 = \kappa_1$, so it must be that $F_h(\hat{S}) \geq \alpha_1$. This implies that $F(\hat{S}) \geq \alpha$, so the basic submodular set cover problem is also solved.

Thus, Definition B.6 is equivalent to the basic submodular set cover problem, and once applied to Definition 4.7 solves the approximate submodular set cover problem. $\square$

Notice that $\epsilon$ can be selected as the smallest distinct difference between values in $F$, and this solves the basic submodular set cover problem exactly.

**Theorem B.8.** *Algorithm 1 using Definition B.6 will approximately solve the non-interactive real-valued submodular set cover with tolerance $\epsilon$ using cost at most $(1 + \ln(\alpha_1')) GCC$.*

*Proof.* Lemma B.7 shows that Definition B.6 can be used to approximately solve the non-interactive real-valued submodular set cover problem.

As discussed in the proof of Theorem 4.6, the upper bound of Algorithm 1 derived in [10] is $\left(1 + \ln(\bar{F}_{max})\right) GCC$. The upper bound in the non-interactive real-valued submodular set cover problem is then:

$$\left(1 + \ln(\bar{F}_{max})\right) GCC = (1 + \ln(|H|C_F C_G)) GCC$$
$$= \left(1 + \ln\left(|H|\alpha_1' D_G^N \prod_{n=1}^{N} (\kappa_n - \kappa_{n-1})\right)\right) GCC$$

$$= \left(1 + \ln\left(\alpha_1'\right)\right) GCC,$$

giving Theorem B.8. □

## C   Analysis of Continous Threshold Curve Version

**Lemma C.1.** $F_h(\hat{S}) \geq \alpha(G_h(\hat{S}))$ *iff* $F_h(\hat{S}) \geq \alpha(\kappa_n)$ *or* $G_h(\hat{S}) \geq \kappa_n$ *for all* $n$.

*Proof.* Suppose that $F_h(\hat{S}) \geq \alpha(G_h(\hat{S}))$. First for all $n$ where $\kappa_n \leq G_h(\hat{S})$, the condition is already satisfied. For the remaining $\kappa_n > G_h(\hat{S})$, notice that $\alpha$ is non-increasing. Then, $\alpha(G_h(\hat{S})) \geq \alpha(\kappa_n)$, and $F_h(\hat{S}) \geq \alpha(\kappa_n)$.

Next, suppose that $F_h(\hat{S}) \geq \alpha(\kappa_n)$ or $G_h(\hat{S}) \geq \kappa_n$ for all $n$. Note that the $\kappa_n$ thresholds take on all possible values of $G_h$, so $F_h(\hat{S}) \geq \alpha(G_h(\hat{S}))$. □

*Proof of Theorem 4.10.* Lemma C.1 implies that solving solving smooth ISSC instantiated with Definition 4.9 is equivalent to solving smooth ISSC with a convex threshold curve.

Theorem A.5 shows that the upper bound of Algorithm 1 is $\left(1 + \ln(\bar{F}_{max})\right) GCC$. Plugging in the value of $\bar{F}_{max}$, the upper bound in the continuous threshold curve version is then:

$$\left(1 + \ln(\bar{F}_{max})\right) GCC = \left(1 + \ln(|H| C_F C_G)\right) GCC = \left(1 + \ln\left(|H| \alpha_1' D_G^N\right)\right) GCC$$

giving Theorem 4.10. □

## D   Alternative Methods of Solving Multiple Thresholds

This section introduces some alternative methods of smooth ISSC and compares the performance of the methods.

### D.1   Satisfy Thresholds One-by-One

Noisy ISSC [11] can be used to satisfy one threshold at a time. This can be extended to solve smooth ISSC by applying the method $N$ times - once for each threshold. The thresholds can be run in any order, but we will consider running the thresholds forward and backwards in this experiment.

### D.2   Alternative Definition

An alternative form of Definition 4.3 can be made, which results in different performance guarantees.

**Definition D.1** (Alternative form of $\bar{F}$ and $\bar{F}_{max}$)**.**

$$\bar{F}_{h,n}(\hat{S}) \triangleq \left(\left(\kappa_n - \kappa_{n-1}\right) - G_{h,\kappa_n,\kappa_{n-1}}(\hat{S})\right) F_{h,\alpha_n,\alpha_{n+1}}(\hat{S}) + G_{h,\kappa_n,\kappa_{n-1}}(\hat{S})(\alpha_n - \alpha_{n+1}),$$

$$\bar{F}_h(\hat{S}) \triangleq C_{\bar{F}} \sum_{n=1}^{N} \left[\bar{F}_{h,n}(\hat{S})\right], \text{ where } C_{\bar{F}} = D_F D_G,$$

$$\bar{F}(\hat{S}) \triangleq \sum_{h \in H} \bar{F}_h(\hat{S}), \ \ \bar{F}_{max} \triangleq |H| D_F D_G \sum_{n=1}^{N} \left[(\alpha_n - \alpha_{n+1})(\kappa_n - \kappa_{n-1})\right].$$

For $\bar{F}$ to be submodular, we also require Condition D.2.

**Condition D.2.** *The sequences* $\langle \alpha_n - \alpha_{n+1} \rangle_{n=1}^{N}$ *and* $\langle \kappa_n - \kappa_{n-1} \rangle_{n=1}^{N}$ *are non-increasing.*

Lemmas A.1 and A.4 and Theorem A.5 hold without modification.

**Lemma D.3** (Alternative form of Lemma A.2)**.** $\bar{F}(\hat{S}) \geq \bar{F}_{max}$ *if and only if* $F_h(\hat{S}) \geq \alpha_n$ *for all* $h$ *such that* $G_h(S^*) < \kappa_n$ *for* $n \in \{1, \ldots, N\}$.

*Proof.* Due to Lemma A.1, it is equivalent to show that $\bar{F}(\hat{S}) \geq \bar{F}_{max}$ if and only if $F_h(\hat{S}) \geq \alpha_n$ for all $h$ such that $G_h(\hat{S}) < \kappa_n$ for $n \in \{1, \ldots, N\}$.

First, suppose that $\bar{F}(\hat{S}) \geq \bar{F}_{max}$. $\bar{F}(\hat{S})$ may not exceed its maximum value, so

$$\bar{F}(\hat{S}) = \bar{F}_{max} = |H|D_F D_G \sum_{n=1}^{N} [(\alpha_n - \alpha_{n+1})(\kappa_n - \kappa_{n-1})].$$

Note that for all $h \in H$, when $\bar{F}_{h_{max}}$ is defined as the maximum value of $\bar{F}_h$,

$$0 \leq \bar{F}_h(\hat{S}) \leq \bar{F}_{h_{max}} = D_F D_G \sum_{n=1}^{N} [(\alpha_n - \alpha_{n+1})(\kappa_n - \kappa_{n-1})].$$

Then, if $\bar{F}(\hat{S}) = \bar{F}_{max}$, then $\bar{F}_h(\hat{S}) = \bar{F}_{h_{max}}$ for all $h \in H$.

Next, when $\bar{F}_{h,n_{max}}$ is defined as the maximum value of $\bar{F}_{h,n}$,

$$0 \leq \bar{F}_{h,n}(\hat{S}) \leq \bar{F}_{h,n_{max}} = (\alpha_n - \alpha_{n+1})(\kappa_n - \kappa_{n-1}).$$

Then, if $\bar{F}_h(\hat{S}) = \bar{F}_{h_{max}}$, then $\bar{F}_{h,n}(\hat{S}) = \bar{F}_{h,n_{max}}$ for all $h \in H$ and all $n \in \{1, 2, \ldots, N\}$.

Finally, if $\bar{F}_{h,n}(\hat{S}) = \bar{F}_{h,n_{max}}$, then $F_{h,\alpha_n,\alpha_{n+1}}(\hat{S}) = \alpha_n - \alpha_{n+1}$ or $G_{h,\kappa_n,\kappa_{n-1}}(\hat{S}) = \kappa_n - \kappa_{n-1}$. If $F_{h,\alpha_n,\alpha_{n+1}}(\hat{S}) = \alpha_n - \alpha_{n+1}$, then $F_h(\hat{S}) \geq \alpha_n$, and if $G_{h,\kappa_n,\kappa_{n-1}}(\hat{S}) = \kappa_n - \kappa_{n-1}$, then $G_h(\hat{S}) \geq \kappa_n$. This implies that $F_h(\hat{S}) \geq \alpha_n$ for all $h$ such that $G_h(\hat{S}) < \kappa_n$ for $n \in \{1, \ldots, N\}$.

For the opposite direction, suppose that $F_h(\hat{S}) \geq \alpha_n$ for all $h$ such that $G_h(\hat{S}) < \kappa_n$ for $n \in \{1, \ldots, N\}$. This means that for all $h \in H$ and all $n \in \{1, \ldots, N\}$, $F_h(\hat{S}) \geq \alpha_n$ or $G_h(\hat{S}) \geq \kappa_n$. Then, $F_{h,\alpha_n,\alpha_{n+1}}(\hat{S}) = (\alpha_n - \alpha_{n+1})$ or $G_{h,\kappa_n,\kappa_{n-1}}(\hat{S}) = (\kappa_n - \kappa_{n-1})$. Then, $\bar{F}_{h,n}(\hat{S}) = (\alpha_n - \alpha_{n+1})(\kappa_n - \kappa_{n-1})$, $\bar{F}_h(\hat{S}) = D_F D_G \sum_{n=1}^{N} [(\alpha_n - \alpha_{n+1})(\kappa_n - \kappa_{n-1})]$, and $\bar{F}(\hat{S}) = |H|D_F D_G \sum_{n=1}^{N} [(\alpha_n - \alpha_{n+1})(\kappa_n - \kappa_{n-1})] = \bar{F}_{max}$. $\square$

**Lemma D.4** (Alternative form of Lemma A.3)**.** *Let $F_h(\hat{S})$ and $G_h(\hat{S})$ be monotone non-decreasing submodular functions, and let the sequences $\langle \alpha_i - \alpha_{n+1} \rangle_{i=1}^{N}$ and $\langle \kappa_n - \kappa_{n-1} \rangle_{i=1}^{N}$ for $n \in \{1, \ldots, N\}$ be non-increasing [Condition D.2]. Then, $\bar{F}(\hat{S})$ from Definition D.1 is a monotone non-decreasing submodular function.*

*Proof.* Define $\delta_S(F, x) \triangleq F(S \oplus x) - F(S)$. First, we show that $\delta_A(\bar{F}, x) \geq 0$ for all $A$ and $x$:

$$\delta_A(\bar{F}, x) = \sum_{h \in H} C_{\bar{F}} \sum_{n=1}^{N} [\delta_A(\bar{F}_{h,n}, x)]$$

$$= \sum_{h \in H} C_{\bar{F}} \sum_{n=1}^{N} \Big[ (\kappa_n - \kappa_{n-1}) \delta_A(F_{h,\alpha_n,\alpha_{n+1}}, x)$$
$$+ \delta_A(G_{h,\kappa_n,\kappa_{n-1}}, x)(\alpha_n - \alpha_{n+1})$$
$$+ F_{h,\alpha_n,\alpha_{n+1}}(A) G_{h,\kappa_n,\kappa_{n-1}}(A)$$
$$- F_{h,\alpha_n,\alpha_{n+1}}(A \oplus x) G_{h,\kappa_n,\kappa_{n-1}}(A \oplus x) \Big]$$

$$= \sum_{h \in H} C_{\bar{F}} \sum_{n=1}^{N} \Big[ ((\kappa_n - \kappa_{n-1}) - G_{h,\kappa_n,\kappa_{n-1}}(A)) \delta_A(F_{h,\alpha_n,\alpha_{n+1}}, x)$$
$$+ \delta_A(G_{h,\kappa_n,\kappa_{n-1}}, x)((\alpha_n - \alpha_{n+1}) - F_{h,\alpha_n,\alpha_{n+1}}(A \oplus x)) \Big]$$

Note that $(\kappa_n - \kappa_{n-1}) - G_{h,\kappa_n,\kappa_{n-1}}(A)$, $\delta_A(F_{h,\alpha_n,\alpha_{n+1}}, x)$, $\delta_A(G_{h,\kappa_n,\kappa_{n-1}}, x)$, and $(\alpha_n - \alpha_{n+1}) - F_{h,\alpha_n,\alpha_{n+1}}(A \oplus x)$ are all non-negative. Thus, $\delta_A(F_h, x) \geq 0$, and $F_h(S)$ is non-decreasing.

Next, consider any $B$ such that $A \subseteq B$. Similarly,

$$\delta_B(\bar{F}, x) \triangleq \sum_{h \in H} C_{\bar{F}} \sum_{n=1}^{N} \Bigg[ ((\kappa_n - \kappa_{n-1}) - G_{h,\kappa_n,\kappa_{n-1}}(B)) \delta_B(F_{h,\alpha_n,\alpha_{n+1}}, x)$$

$$+ \delta_B(G_{h,\kappa_n,\kappa_{n-1}}, x)((\alpha_n - \alpha_{n+1}) - F_{h,\alpha_n,\alpha_{n+1}}(B \oplus x)) \Bigg]$$

Then, we show that for all $A \subseteq B$, $\delta_B(\bar{F}, x) - \delta_A(\bar{F}, x) \geq 0$.

$$\delta_B(\bar{F}, x) - \delta_A(\bar{F}, x) \triangleq \sum_{h \in H} C_{\bar{F}} \sum_{n=1}^{N} \Bigg[ (\kappa_n - \kappa_{n-1})(\delta_B(F_{h,\alpha_n,\alpha_{n+1}}, x) - \delta_A(F_{h,\alpha_n,\alpha_{n+1}}, x))$$

$$- G_{h,\kappa_n,\kappa_{n-1}}(B)\delta_B(F_{h,\alpha_n,\alpha_{n+1}}, x)$$

$$+ G_{h,\kappa_n,\kappa_{n-1}}(A)\delta_A(F_{h,\alpha_n,\alpha_{n+1}}, x)$$

$$+ (\alpha_n - \alpha_{n+1})(\delta_B(G_{h,\kappa_n,\kappa_{n-1}}, x) - \delta_A(G_{h,\kappa_n,\kappa_{n-1}}, x))$$

$$- \delta_B(G_{h,\kappa_n,\kappa_{n-1}}, x)F_{h,\alpha_n,\alpha_{n+1}}(B \oplus x)$$

$$+ \delta_A(G_{h,\kappa_n,\kappa_{n-1}}, x)F_{h,\alpha_n,\alpha_{n+1}}(A \oplus x) \Bigg]$$

Note that $G_{h,\kappa_n,\kappa_{n+1}}(A) \leq G_{h,\kappa_n,\kappa_{n+1}}(B)$ and $F_{h,\alpha_n,\alpha_{n+1}}(A \oplus x) \leq F_{h,\alpha_n,\alpha_{n+1}}(B \oplus x)$. Then, $\delta_B(\bar{F}, x) - \delta_A(\bar{F}, x) \leq \sum_{h \in H} C_{\bar{F}} \mathbb{L}_h$, where,

$$\mathbb{L}_h = \sum_{n=1}^{N} \Bigg[ (\kappa_n - \kappa_{n-1})(\delta_B(F_{h,\alpha_n,\alpha_{n+1}}, x) - \delta_A(F_{h,\alpha_n,\alpha_{n+1}}, x))$$

$$- G_{h,\kappa_n,\kappa_{n-1}}(B)\delta_B(F_{h,\alpha_n,\alpha_{n+1}}, x) + G_{h,\kappa_n,\kappa_{n-1}}(B)\delta_A(F_{h,\alpha_n,\alpha_{n+1}}, x)$$

$$+ (\alpha_n - \alpha_{n+1})(\delta_B(G_{h,\kappa_n,\kappa_{n-1}}, x) - \delta_A(G_{h,\kappa_n,\kappa_{n-1}}, x))$$

$$- \delta_B(G_{h,\kappa_n,\kappa_{n-1}}, x)F_{h,\alpha_n,\alpha_{n+1}}(B \oplus x) + \delta_A(G_{h,\kappa_n,\kappa_{n-1}}, x)F_{h,\alpha_n,\alpha_{n+1}}(B \oplus x) \Bigg]$$

$$= \sum_{n=1}^{N} \Bigg[ ((\kappa_n - \kappa_{n-1}) - G_{h,\kappa_n,\kappa_{n-1}}(B))(\delta_B(F_{h,\alpha_n,\alpha_{n+1}}, x) - \delta_A(F_{h,\alpha_n,\alpha_{n+1}}, x))$$

$$+ ((\alpha_n - \alpha_{n+1}) - F_{h,\alpha_n,\alpha_{n+1}}(B \oplus x))(\delta_B(G_{h,\kappa_n,\kappa_{n-1}}, x) - \delta_A(G_{h,\kappa_n,\kappa_{n-1}}, x)) \Bigg]$$

$$= \Bigg[ \sum_{n=1}^{N} ((\kappa_n - \kappa_{n-1}) - G_{h,\kappa_n,\kappa_{n-1}}(B))(\delta_B(F_{h,\alpha_n,\alpha_{n+1}}, x) - \delta_A(F_{h,\alpha_n,\alpha_{n+1}}, x))$$

$$+ \sum_{n=1}^{N} ((\alpha_n - \alpha_{n+1}) - F_{h,\alpha_n,\alpha_{n+1}}(B \oplus x))(\delta_B(G_{h,\kappa_n,\kappa_{n-1}}, x) - \delta_A(G_{h,\kappa_n,\kappa_{n-1}}, x)) \Bigg].$$

Note that the sequence $(\kappa_n - \kappa_{n-1}) - G_{h,\kappa_n,\kappa_{n-1}}(B)$ must take the form $\langle 0, \ldots, 0, a, \kappa_{n+1} - \kappa_n, \ldots, \kappa_N - \kappa_{N-1} \rangle$ where $a \in [0, \kappa_n - \kappa_{n-1}]$. Because of the restriction on the values of $\kappa_n - \kappa_{n-1}$ (Condition D.2), this sequence is non-increasing. In addition, $\sum_{n=j}^{N} \delta_A(F_{h,\alpha_n,\alpha_{n+1}}, x) \geq \sum_{n=1}^{j} \delta_B(F_{h,\alpha_n,\alpha_{n+1}}, x)$ for all positive integer $j \leq N$. Thus, $\sum_{n=1}^{N} ((\kappa_n - \kappa_{n-1}) - G_{h,\kappa_n,\kappa_{n-1}}(B))(\delta_B(F_{h,\alpha_n,\alpha_{n+1}}, x) - \delta_A(F_{h,\alpha_n,\alpha_{n+1}}, x))$ is non-positive.

Note also that the sequence $(\alpha_n - \alpha_{n+1}) - F_{h,\alpha_n,\alpha_{n+1}}(B \oplus x)$ must take the form $\langle \alpha_1 - \alpha_2, \ldots, \alpha_{n-1} - \alpha_n, a, 0, \ldots, 0 \rangle$ where $a \in [0, \alpha_n - \alpha_{n+1}]$. Because of the restriction on the values of $\alpha_n - \alpha_{n+1}$ (Condition D.2), this sequence is non-increasing. In addition, $\sum_{n=1}^{j} \delta_A(G_{h,\kappa_n,\kappa_{n-1}}, x) \geq \sum_{n=1}^{j} \delta_B(G_{h,\kappa_n,\kappa_{n-1}}, x)$ for all positive integer $j \leq N$. Thus, $\sum_{n=1}^{N} ((\alpha_n - \alpha_{n+1}) - F_{h,\alpha_n,\alpha_{n+1}}(B \oplus x))(\delta_B(G_{h,\kappa_n,\kappa_{n-1}}, x) - \delta_A(G_{h,\kappa_n,\kappa_{n-1}}, x))$ is non-positive.

These two statements imply that $\delta_B(F, x) - \delta_A(F, x) \leq 0$, which means that $F$ is submodular. $\quad\square$

**Theorem D.5** (Alternative form of Theorem 4.6). *Given Condition 4.5, Algorithm 1 using Definition 4.4 solves the multiple thresholds version of Problem 1 using cost at most* $\left(1 + \ln\left(|H| D_F D_G \sum_{n=1}^{N} [(\alpha_n - \alpha_{n+1})(\kappa_n - \kappa_{n-1})]\right)\right) GCC.$

*Proof.* Lemma A.2 implies that satisfying the condition $\bar{F} \geq \bar{F}_{max}$ is equivalent to satisfying the goal of smooth ISSC with multiple thresholds. Next, Lemma D.4 implies that $\bar{F}$ may be used with Algorithm 1 and have guaranteed performance bounds. Finally, Theorem A.5 shows that the upper bound of Algorithm 1 is $\left(1 + \ln(\bar{F}_{max})\right) GCC$. Plugging in the value of $\bar{F}_{max}$, the upper bound in the multiple threshold case of smooth ISSC is then:

$$\left(1 + \ln(\bar{F}_{max})\right) GCC = \left(1 + \ln\left(|H|D_F D_G \sum_{n=1}^{N} \left[(\alpha_n - \alpha_{n+1})(\kappa_n - \kappa_{n-1})\right]\right)\right) GCC,$$

giving Theorem D.5. $\qquad\square$

### D.3 Multi-Threshold with Monotone Circuits of Constraints

A simple method of reducing any monotone boolean circuit of constraints to a single constraint is introduced in [11]. To do so, they show that the OR of two constraints $(\hat{F}_i(S) \geq \alpha_i) \vee (\hat{F}_j(S) \geq \alpha_j)$ can be reduced to a single constraint $\bar{F}(S) = \bar{\alpha}$ with $\bar{F}(S) \triangleq (\alpha_i - \min(\hat{F}_i(S), \alpha_i)) \min(\hat{F}_j(S), \alpha_j) + \min(\hat{F}_i(S), \alpha_i)\alpha_j$ and $\bar{\alpha} = \alpha_i\alpha_j$, and they show that the AND of two constraints $(\hat{F}_i(S) \geq \alpha_i) \wedge (\hat{F}_j(S) \geq \alpha_j)$ can be reduced to a single constraint $\bar{F}(S) = \bar{\alpha}$ with $\bar{F}(S) \triangleq \min(\hat{F}_i(S), \alpha_i) + \min(\hat{F}_j(S), \alpha_j)$ and $\bar{\alpha} = \alpha_i + \alpha_j$.

Note that Figure 3 expresses smooth ISSC as a monotone boolean circuit of constraints. Thus, the reduction method from [11] is directly applicable to smooth ISSC. The application of this reduction method results in Definition D.6.

**Definition D.6** ($\bar{F}$ and $\bar{F}_{max}$ from Direct Reduction of a Monotone Boolean Circuit of Constraints)**.**

$$\bar{F}_{h,n}(\hat{S}) \triangleq \left(\kappa_n - \min(G_h(\hat{S}), \kappa_n)\right) \min(F_h(\hat{S}), \alpha_n) + \min(G_h(\hat{S}), \kappa_n)\alpha_n,$$

$$\bar{F}_h(\hat{S}) \triangleq C_{\bar{F}} \sum_{n=1}^{N} \bar{F}_{h,n}(\hat{S}), \quad C_{\bar{F}} = D_F D_G,$$

$$\bar{F}(\hat{S}) \triangleq \sum_{h \in H} \bar{F}_h(\hat{S}), \quad \bar{F}_{max} \triangleq |H|D_F D_G \sum_{n=1}^{N} \alpha_n \kappa_n$$

This definition no longer has to satisfy Condition 4.5 or Condition D.2.

**Theorem D.7.** *Algorithm 1 using Definition D.6 solves the multiple thresholds version of Problem 1 using cost at most* $\left(1 + \ln\left(|H|D_F D_G \sum_{n=1}^{N} \alpha_n \kappa_n\right)\right) GCC$.

*Proof.* Definition D.6 is equivalent to the original problem because it is reducing a ciruit of AND and OR constraints. Theorem A.5 shows that the upper bound of Algorithm 1 is $\left(1 + \ln(\bar{F}_{max})\right) GCC$. Plugging in the value of $\bar{F}_{max}$, the upper bound in the multiple threshold case of smooth ISSC is then:

$$\left(1 + \ln(\bar{F}_{max})\right) GCC = \left(1 + \ln\left(|H|D_F D_G \sum_{n=1}^{N} \alpha_n \kappa_n\right)\right) GCC,$$

giving Theorem D.5. $\qquad\square$

Notice that the approximation bound in this formulation is not strictly better or worse than the approximation bound from Definition 4.4. Depending on the choice of thresholds, either formulation can have a better approximation bound. In contrast, the approximation bound in this formulation is never better than the approximation bound from Definition D.1.

## E   Details of Empirical Validation

### E.1   Comparison of Methods to Solve Multiple Thresholds

This section describes the three settings from Section 5 used to compare the performance of our method from Section 4.2 with the methods from Section D.

In setting A, we constructed 100 hypotheses and 1000 queries. Each hypothesis is given a 50% probability of responding "yes" when given a query, and a 50% chance of responding "no" when given a query. In addition, each hypothesis-query pair given random utility of 0 or 1 and a random distance of 0 or 1. The utility function $F_h$ for each hypothesis is then defined as the sum of the utilities for queries that it responded "yes" to, and the distance function $G_h$ for each hypothesis is defined as the sum of the distances for queries where it's response does not match the response of the target hypothesis. The thresholds used were $\alpha_i = \{15, 14, \ldots, 2, 1\}$ and $\kappa_i = \{1, 2, \ldots, 14, 15\}$.

In setting B, we use a non-interactive setting where only one possible response is available. We constructed 30 hypotheses and 400 queries divided into two groups of 200 denoted as $X$ queries and $Y$ queries. Each query is assigned to 3 random hypotheses. The presence of an $X$ queries assigned to a hypothesis will set $F_h$ to 10, but extra $X$ queries will not increase $F_h$ further. In addition, $F_h$ is increased by 6 for each $Y$ query assigned to the corresponding hypothesis. $G_h$ is not affected by $X$ queries, and is increased by 6 for each $Y$ query assigned to the corresponding hypothesis. The thresholds used where $\alpha_i = \{20, 10\}$ and $\kappa_i = \{10, 20\}$.

Setting C is identical to setting B, except that $F_h$ and $G_h$ are swapped.

In Figure 4, we compare the costs of several different methods. In this figure, the costs from different random instantiations of the experiment described above are sorted and then plotted in increasing order.

| Method | Setting A | Settings B and C |
|---|---|---|
| Multiple Threshold (Definition 4.4) | 1500 | 60000 |
| Alternative (Definition D.1) | 1500 | 6000 |
| Circuit of Constraints (Definition D.6) | 68000 | 12000 |

Table 1: $\bar{F}_{max}$ for the methods tested in Section 5.

Table E.1 shows $\bar{F}_{max}$ for the different methods. Notice that the approximation guarantee is worst for the circuit of constraints method in setting A, where it had the best performance, and the approximation guarantee is worst for the multiple thresholds method in Settings B and C, where it was tied for the best performing method. This indicates that the worst-case guarantee is not a reliable estimate of the actual performance.

### E.2 Validating Approximation Tolerances

This section describes the experimental setup from Section 5. The hypotheses are denoted $H = \{h_1, h_2, \ldots, h_{50}\}$. The queries are denoted $\mathcal{Q} = \{q_{i,j} | i \in \{1, 2, \ldots, 50\}, j \in \{1, 2, \ldots, 10\}\}$. The responses are "yes" and "no", representing interest in the query by the hypothesis. Each cluster was assigned 25 random hypotheses that are interested in it, and the target hypothesis was assigned 25 random clusters to be interested in. Let $c_{h_i}$ be the set of clusters that hypothesis $h_i$ is interested in.

Let $c_i(\hat{S}) = \{q_{i,j} | q_{i,j} \in \hat{S}, j \in \{1, 2, \ldots, 10\}\}$. This is the set of queries that has been recommended and is in cluster $i$. The utility functions are defined as the following:

$$F_{h_i}(\hat{S}) = \sum_{j=c_{h_i}} \sqrt{|c_j(\hat{S})|}$$

This definition of utility increases the hypothesis gets more queries it is interested in. However, redundant queries from the same cluster are given diminished weight. The distance functions $G_h$ are defined as the number of responses obtained that are different for the hypothesis being considered and the target hypothesis. The thresholds used were $\alpha_i = \{15, 14, \ldots, 2, 1\}$ and $\kappa_i = \{1, 2, \ldots, 14, 15\}$.