[Reviews · NeurIPS 2015]

Submitted by Assigned_Reviewer_1

*Summary*

This paper considers the interactive submodular set cover problem, where one is given a set of submodular functions, and needs to optimize an unknown true function using a minimal amount of actions. The paper considers two extensions of previous work: (1) varying "saturation" conditions (i.e., threshold) for different hypothesis functions in the hypothesis class; and (2) settings when the hypothesis submodular functions are real-valued. The paper provides a unified greedy framework to (approximately) solve both problems. In general the paper is well organized and the theory seems sound, even though some parts could have been made clearer. My major criticism is the empirical part of the paper, namely there is no empirical support why one should use the proposed algorithm. please see my detailed comments below.

*Quality*

The paper provides worst case cost bound for the smooth interactive set cover problem, by designing a submodular meta-objective objective for various scenarios (of the threshold function). The key result of the paper the approximation guarantee for the multiple threshold version (and the authors show that one can reduce the convex threshold curve version can be approximately reduced to the previous case). The theory and proofs non-trivially generalize the results of Guillory and Bilmes, 2010.

The authors gives an example of a real-world applications, but show no empirical results. In the simulation results, there are no competitive approaches (even random selection) as well, thus it's hard to tell the empirical effectiveness of the algorithm.

The bound depends on the number of thresholds $N$. When N=1 it reduces to the Noisy ISSC setting. It would be helpful to see empirically how different N's influence the performance of the algorithm.

*Clarity*

The paper mentions active learning a couple of times in the paper. However, along the line of submodular maximization, active learning is more of "exploring" tests that can mostly reduces size of the version space, and does not favors the actions that are "exploitative". It may cause some confusion to readers.

There are a list of typos / points that are unclear, as summarized below.

Ln 73, in the cost bound, define $a$. Ln 208-210, if alpha_n > alpha_j, then by Ln 140, it should hold that kapa_n < kapa_j. Hence the kapa_n and kapa_j in equation (2) should be flipped.

Ln 230, from the figure, shouldn't the condition only enforces that either of the two threshold must be satisfied, according to (A) (Ln 228-229)?

In section 4.5, there seem to be a couple of typos / unclear explanations to the experiment: Ln 327, each cluster was randomly assigned 15 clusters of interest: do you mean 15 hypotheses of interest? Ln 334 states that "at epsilon = 15, the algorithm is allowed to terminate immediate immediately". Why is the cost still non-negative when epsilon >= 15?

Ln 429, threshold for to

*Originality and Significance*

The work builds upon and extends the work of Guillory and Bilmes, 2010. As I mentioned earlier, even though it relies heavily on the previously work, the generalization in the proofs is novel.

Since the construction of the algorithm (specifically, the surrogate meta objective) is non-trivial, I would like to see some empirical evidences showing that why one should use the proposed approach.
Summary: The paper proposes a non-trivial generalization to the interactive submodular set cover problem. The theory and proofs in the paper seem to be sound. There are a few clarity issues, and the empirical analysis is relatively weak.

Submitted by Assigned_Reviewer_2

The paper proposes a greedy algorithm for smooth interactive submodular set cover, where we have a set of hypotheses and one can smoothly vary the threshold to optimize more the most plausible hypothesis. It also extends the analysis to real valued submodular functions.

Some comments: \espilon is defined as a fixed value in section 3, but it doesn't seem to be constant in Figure 1, d? Should it also be fixed in the figure?

The experimental section, in particular, is much too terse and there is no baseline comparison. Even if the proposed method is the first one for satisfying a set of plausible submodular functions, a simple base line could be trying to satisfy the functions one by one, and compare the costs in Figure 3.

It's not clear what functions/values has been used for \alpha and \kappa in the experiments.

Also, there is no plot showing the cost and running time of the algorithm for different \alpha and \kappa s. One potential choice would be simple linear functions for both of the parameters.

In section 4.5, it has mentioned that for \epsilon=15, the algorithm is allowed to terminate immediately. Why is that?

Moreover, there is no experiments showing the performance of the algorithm on real-valued submodular functions.

The computation complexity of the algorithms hasn't been discussed anywhere in the paper.

Can the authors elaborate on how does \bar{F} makes a trade off between exploration and exploitation?

Finally, in Definition 4.4 has been wrongly referred to in page 5. Also, in page 4, Lemma 4.2 corresponds to Lemma 2 in [11].
Summary: The paper proposes a greedy algorithm for smooth interactive submodular set cover. Overall, I think the paper does a good job of providing theoretical guarantees for the proposed algorithms, but experiments are too limited to show the quality of the solution.

Submitted by Assigned_Reviewer_3

This paper considers a generalization of the Interactive Submodular Set Cover (ISSC) problem. In ISSC, the goal is to interactively collect elements until the value of the set of elements, represented by an unknown submodular function, reaches some threshold. In the original ISSC there is a single correct submodular function, which can be revealed using responses to each selected element, and a single desired threshold. This paper proposes to simultaneously require reaching some threshold for all the possible submodular functions. The threshold value is determined as a convex function of a submodular agreement measure between the given function and the responses to all elements. Each element has a cost, and so the goal is to efficiently decide which elements to collect to satisfy the goal at a small cost.

The main contribution of this work is to propose a greedy algorithm based on a single submodular function. The results guarantee that this algorithm pays a cost which is not more than a factor more than the optimal cost. The factor is linear in the number of different thresholds.

The paper is overall well presented and the problem it attempts to solve could be of interest. The motivation, however, is not clearly spelled out. What are interesting cases where this goal of multiple thresholds or a convex function make sense?

The cost approximation factor that the authors show is linear in the number of thresholds. Thus it remains open whether a simpler algorithm, that collects elements for each threshold individually would not achieve similar results.

Summary: The problem is interesting, but not very well motivated. The results are reasonable although not very strong.

Author Feedback
Author rebuttal: We thank the reviewers for their insightful comments! We first make some global comments, and then respond to other specific issues raised by individual reviewers.

*** GLOBAL COMMENTS ***

>> Main Contribution

We emphasize that our main contribution is a theoretical one. First, our generalization of noisy ISSC is highly non-trivial. Second, our solution concept & approach to approximate real-valued submodular set cover are novel in their own right, especially given the paucity of prior results for this setting (our literature review did not uncover any prior results on real-valued submodular set cover). Third, the two theoretical contributions naturally complement each other since smooth ISSC naturally yields real-valued submodular set cover problems.

In retrospect, we realize that our paper, as written, does not properly highlight the technical insights of our theoretical contributions, but rather over-focuses on connecting to practical applications of submodular optimization. If accepted, we will re-focus the exposition to be more clear in its mission.

>> Practicality & Motivation

While our contributions are primarily theoretical, our approach is motivated by general practical considerations. One important limitation of noisy ISSC by Guillory & Bilmes is its "all or nothing" nature of having a single alpha/kappa target threshold. Most motivating applications considered in Guillory & Bilmes could be better modeled using more flexible target criteria. One example is the content recommendation setting described in Sec. 2 that was also discussed in Guillory & Bilmes.

>> Empirical Results

We have begun empirically validating our approach against several heuristic baselines, such as randomly or intelligently choosing a single threshold at a time. Our preliminary results suggest that our approach is always competitive with all baselines, and is sometimes significantly better. This is because all natural baselines that we considered have much weaker worse-case guarantees, whereas our approach is principled and will always perform reasonably well. If accepted, we would be happy to add more empirical comparisons vs baselines to the paper.

*** RESPONSES TO OTHER SPECIFIC COMMENTS/QUESTIONS ***
(Due to space constraints, we could not respond to all comments.)

>> R1, regarding Ln 208-210:

When the doubly truncated version of G is used, the arguments are put in the opposite order as F. For example, on line 236, the arguments for G are ordered kappa_{n} and kappa_{n-1}, while those for F are ordered alpha_{n} and alpha_{n+1}. For both, the larger argument comes 1st (corresponding to n), and the smaller argument comes 2nd (corresponding to j).

>> R1, regarding Ln 230:

Each threshold refers to a pair alpha_i/kappa_i collectively.

>> R1, regarding Ln 327:

This should be "Each hypothesis was randomly assigned 15 clusters of interest".

>> R1, regarding Ln 334:

The cost is always non-negative. In the experiments, cost is non-zero because our approximation guarantee is fairly conservative. Even though the algorithm is allowed to terminate immediately, it actually comes very close to satisfying the thresholds.

>> R2, regarding Figure 1d:

Figure 1d is an optical illusion, and epsilon is actually constant.

>> R2, regarding values of alpha & kappa in experiments:

The following values were used:
alpha = 15, 14, ..., 2, 1
kappa = 1, 2, ..., 14, 15

>> R2, regarding immediate termination for epsilon=15:

Epsilon represents the amount that F is allowed to fall short of the target alpha threshold. Because the largest alpha is 15, when epsilon = 15, F=0 and is already within the tolerance.

>> R2, regarding experiments on real-valued submodular functions:

The experiments in Sec 4.5 evaluate the cost and deviation for real-valued submodular set cover.

>> R2, regarding computational complexity:

The surrogate \bar{F} does not need to be computed for all values, but rather as needed for the greedy algorithm. Thus, our method has a similar runtime as a standard greedy algorithm, with the exception that computing \bar{F} adds a small overhead.

>> R2, elaborate on \bar{F} for explore/exploit:

Figure 2, and the paragraph below Definition 4.4 describes in detail the explore/exploit tradeoff.

>> R3, whether simpler algorithms can achieve similar results:

All simpler algorithms that we considered have much weaker worst-case guarantees than our method. We can add a discussion of this to the paper.

>> R3, regarding proof of Lemma A.1:

This is a typo and is corrected by changing the second and third >= to < on line 498.

>> R3, regarding where GCC or Lemma 4.2 are used:

The worst-case cost is defined as the maximum cost under the specified question asking strategy (over the hypothesis class). Lemma 4.2 shows that GCC >= OPT. All of our results give bounds of the form O(GCC*log(F_max)), which combined with Lemma 4.2 gives a bound vs OPT.